# Inhibition of Protein Aggregation and Endoplasmic Reticulum Stress as a Targeted Therapy for α-Synucleinopathy

**DOI:** 10.3390/pharmaceutics15082051

**Published:** 2023-07-30

**Authors:** Natalia Siwecka, Kamil Saramowicz, Grzegorz Galita, Wioletta Rozpędek-Kamińska, Ireneusz Majsterek

**Affiliations:** Department of Clinical Chemistry and Biochemistry, Medical University of Lodz, 92-215 Lodz, Poland; natalia.siwecka@stud.umed.lodz.pl (N.S.); kamil.saramowicz@stud.umed.lodz.pl (K.S.); grzegorz.galita@umed.lodz.pl (G.G.); wioletta.rozpedek@umed.lodz.pl (W.R.-K.)

**Keywords:** α-synuclein, α-synucleinopathy, Parkinson’s disease, oligomers, fibrils, protein aggregates, Lewy bodies, ER stress, small-molecule inhibitors, disease-modifying strategies

## Abstract

α-synuclein (α-syn) is an intrinsically disordered protein abundant in the central nervous system. Physiologically, the protein regulates vesicle trafficking and neurotransmitter release in the presynaptic terminals. Pathologies related to misfolding and aggregation of α-syn are referred to as α-synucleinopathies, and they constitute a frequent cause of neurodegeneration. The most common α-synucleinopathy, Parkinson’s disease (PD), is caused by abnormal accumulation of α-syn in the dopaminergic neurons of the midbrain. This results in protein overload, activation of endoplasmic reticulum (ER) stress, and, ultimately, neural cell apoptosis and neurodegeneration. To date, the available treatment options for PD are only symptomatic and rely on dopamine replacement therapy or palliative surgery. As the prevalence of PD has skyrocketed in recent years, there is a pending issue for development of new disease-modifying strategies. These include anti-aggregative agents that target α-syn directly (gene therapy, small molecules and immunization), indirectly (modulators of ER stress, oxidative stress and clearance pathways) or combine both actions (natural compounds). Herein, we provide an overview on the characteristic features of the structure and pathogenic mechanisms of α-syn that could be targeted with novel molecular-based therapies.

## 1. Introduction

Parkinson’s disease (PD) is the second most common neurodegenerative disease after Alzheimer’s disease and the most common movement disorder [1,2]. PD is also regarded as the fastest growing neurological disease, affecting over 10 million individuals worldwide [3]. The global prevalence of PD is about 1% of the population over the age of 60, having risen by 74.3% from 1990 to 2016 [4]. In the United States alone, PD was estimated to have an economic impact of USD 52 billion per year [5]. As PD remains incurable, with the symptoms irreversibly deteriorating with the disease progression, it is of utmost importance to improve PD patients’ quality of life. The major molecular event underlying the pathophysiology of PD is abnormal accumulation of α-synuclein (α-syn) in the substantia nigra (SN) of the midbrain, which contributes to progressive loss of dopaminergic neurons [6]. As the disease progresses, α-syn inclusions affect other types of neurons and glial cells and disperse into other brain regions [7]. The diseases that share the common pathological hallmark of α-syn inclusions in the central and peripheral nervous system (CNS; PNS) are known under the umbrella term ‘α-synucleinopathy’ [8]. PD is the most frequent α-synucleinopathy, whereas the other known forms of these disorders comprise, among others, dementia with Lewy bodies (DLB), multiple system atrophy (MSA), PD with dementia and neuroaxonal dystrophy [9]. The mentioned diseases differ in terms of cellular and anatomical localization of α-syn aggregates and, as a consequence, present distinct clinical features and progression dynamics. For instance, in DLB, α-syn co-accumulates with amyloid β (Aβ) and tau protein in neurons and astrocytes, mainly in the cerebral cortex, hippocampus and limbic system, and, in MSA, oligodendrocytes are loaded with α-syn inclusions in various brain regions; also, both conditions progress more rapidly than PD [10,11,12,13]. Until lately, it was unclear how a single protein is able to cause such a variety of different diseases. It has recently been found that the types of α-syn strains significantly vary in different α-synucleinopathies [14,15]. Therefore, pathologies related to α-synuclein are highly heterogenous and remain a challenge in designing effective treatment strategies.

Physiologically, α-syn presents as a monomer that binds to lipid membranes, where it regulates synaptic vesicle trafficking [16]. Numerous factors, including point mutations or multiplications of *SNCA* gene encoding for α-syn, post-translational modifications (PTMs), oxidative stress and aging, may disrupt the balance between α-syn production and clearance and thus make it prone to aggregate [17,18,19,20]. A-syn may misfold into various forms, from highly heterogeneous oligomers through protofibrils to mature fibrils, which eventually assemble into Lewy bodies (LBs) [21]. Among all forms of α-syn, the oligomeric species are considered the most relevant as they may exert neurotoxicity in a variety of mechanisms—proteasome inhibition, lysosome and mitochondrial dysfunction, endoplasmic reticulum (ER) stress, neuroinflammation and ultimately apoptosis [22,23,24]. At the molecular level, oligomers bind to lipids and increase permeability of intracellular membranes, which leads to calcium influx, ion homeostasis dysregulation and caspase-3-dependent cell death [25]. Moreover, failure of the intracellular clearance pathways like the ubiquitin–proteasome system (UPS) or autophagy may lead to the pathological release of α-syn oligomers to the extracellular space (ECS), where they impair synaptic function, self-replicate and further propagate from cell to cell in prion-like fashion [26,27]. Notably, the oligomers were demonstrated to progressively spread to the other regions of the brain, affect different cell types and accumulate in the PNS or even in the transplanted embryonic cells via ‘host-to-the-graft’ transmission [27]. This has been confirmed in several xenograft models of PD, in which patient-derived brain homogenates inoculated into the brain of model animals induced α-syn pathology [14,28]. On the other hand, LBs seem to be in fact protective and represent a form of aggresome [29]; hence, the most optimal approach would be to target the toxic oligomers. The exact molecular mechanisms in which α-syn contributes to dopaminergic neuron loss and disease progression have been reviewed elsewhere [30,31].

Currently, the only available treatment options for PD are symptomatic, specifically in terms of dopamine (DA) replacement therapy [32]. This approach may, however, facilitate the disease progression by promoting formation of toxic oligomers, given the prooxidant properties of DA [33]. Other available medications for PD include, among others, dopamine agonists, monoamine oxidase B (MAO B) and catechol-O-methyl transferase (COMT) inhibitors, anticholinergics and amantadine. Non-pharmacological approaches involve physiotherapy as well as surgical procedures (mainly deep brain stimulation—DBS) [34]. There is no available neuroprotective or neurorestorative therapy for PD to date, although multiple attempts have been undertaken to develop disease-modifying strategies—especially the α-syn-targeting ones. Such attempts focus on different aspects of the α-syn aggregation process, including reducing the production, increasing the clearance, monomer stabilization, disruption of aggregated forms and preventing the formation or propagation of the toxic aggregated species. Other therapies based on pathophysiology of the disease have also been proposed, which focus on the molecular targets indirectly implicated in α-syn-induced toxicity. The currently tested α-syn anti-aggregation strategies comprise gene therapy, vaccines, antibodies, molecular chaperones and small molecules. As protein-based therapeutics might have difficulty in passing through the blood–brain barrier (BBB) and trigger collateral immunological reactions, small molecules seem to be more promising candidates for targeting brain diseases [35].

Considering all the above-mentioned aspects, we herein provide an updated overview of novel disease-modifying strategies against α-synucleinopathy under investigation, with a particular focus on pharmacotherapeutic approaches directly or indirectly targeting α-syn. In this review, we aimed to critically analyze each of the proposed therapies regarding advantages, disadvantages and chances for potential clinical application. We have also provided insight into the structure and aggregation process of α-syn, which could be helpful in terms of identification of new therapeutic targets and future drug design. The review is mostly concentrated on PD treatment as most of the papers address this topic, but it does not exclude new therapeutic possibilities for other disease entities that share this common pathological mechanism.

## 2. The Characteristic Features and Roles of Various Strains of α-Synuclein

### 2.1. The Structure of SNCA Gene and α-Synuclein Protein

*SNCA* gene, encoding for α-syn, is located on chromosome 4q21.3-22 [36]. The gene contains seven exons, of which five are protein-coding [37]. The transcription of SNCA is regulated by the zinc finger proteins ZSCAN21 and ZNF219, GATA2 transcription factor, nerve and basic fibroblast growth factors (NGF; bFGF) as well as methylation and microRNAs (miRNAs) [38,39,40,41]. α-syn is a small 14 kDa/140-amino-acid (aa) protein, which belongs to the synuclein family, along with β-synuclein (β-syn) and γ-synuclein (γ-syn) [37]. All family members are characterized by a highly conserved (with a sequence identity ~90%), repetitive α-helical lipid-binding motif in the N-terminus, reminiscent of the lipid-binding motifs of the exchangeable apolipoproteins (Apo). The α- and β-syn are primarily expressed in the presynaptic terminals in the brain tissue, whereas γ-syn is mainly found in the retina and PNS [42].

α-syn is composed of three distinct domains: an amphipathic N-terminus (1–60), a hydrophobic central nonamyloid component (NAC) (61–95) and a hydrophilic C-terminus (96–140) [43]. An essential sequence of 11 residues is repeated seven times throughout the N-terminal and NAC domains. The N-terminus contains four of seven imperfect 11-residue repeats, based on the core consensus sequence of “KTKEGV” [44]. This lysine-rich, hexameric motif mediates a shift in the protein conformation from the primarily disordered structure in the solution to the amphipathic, α-helix secondary structure upon protein–lipid interaction [43]. This enables insertion of α-helical domains into phospholipid membranes and vesicles, which influences their curvature [44]. Moreover, the two sequences in the N-terminus (36–42 and 45–57) are known to regulate monomeric α-syn interactions prior to cross-β formation [45]. The central NAC region is highly hydrophobic and was first isolated as one of the components of amyloid plaques in Alzheimer’s disease patients [46]. It has been found that it is the 12-aa motif at 71–82 residues (“VTGVTAVAQKTV”) that is responsible for the cross-β-sheet folding, aggregation and fibrillation process [47]. Upon fibrillization, the region arranges into a Greek-key motif at the fibril core [48]. Deletion of the region greatly diminished oligomerization and fibrillation of α-syn in experimental conditions [49]. Unlike N-terminal and NAC regions, the C-terminus greatly varies in size and sequence between species [37]. The C-terminal end is rich in proline and acidic aa residues (aspartate and glutamate), which give it high negative charge density and increase polarity [50]; this also acts as an aggregation inhibition region by providing α-syn with electrostatic repulsions [51]. The C-terminal domain is unstructured and flexible, and it regulates the nuclear localization of α-syn as well as provides it chaperone activity by enabling interactions with other proteins, small molecules and metals [52].

The N-terminus is the major site for α-syn point mutations, which are associated with PD, DLB and MSA (namely A30P, A53E, A53T, E46K, G51D and H50Q) [53]. A30P and A53T mutations enhance α-syn oligomerization, whereas E46K, G51D, A53E and H50Q increase the level of fibrillar aggregates and enhance their toxicity [54,55,56,57]. On the other hand, phosphorylation and truncation predominantly occur at the C-terminal end. This disordered domain contains a serine (129) and three tyrosine residues (125, 133 and 136) that are prone to phosphorylation, which may negatively impact physicochemical properties of the protein [58]. Also, truncation of this region was shown to exacerbate aggregation propensity and toxicity of α-syn [59]. Both C-terminal phosphorylated and truncated forms of α-syn have been found in increased amounts in LBs of PD patients [60].

The above-mentioned six missense mutations in *SNCA* as well as multiplications of the gene are well-known risk factors for the autosomal dominant (AD) form of PD. Interestingly, the number of repeats positively correlates with the α-syn level, the age-at-onset and the intensity of symptoms [61]. In that way, increased level of the wild-type protein alone is sufficient to induce the disease. On the other hand, single nucleotide polymorphisms (SNPs) as well as reduced epigenetic silencing of *SNCA* are major risk factors for developing PD [62,63]. For instance, *Rep1* polymorphism in the promoter region of *SNCA* might increase α-syn expression [64]. Based on these findings, it has been suggested that decreasing α-syn expression level may reduce the risk of adopting abnormal conformation by the protein and oligomerization. Although *SNCA* silencing is not lethal, loss of endogenous α-syn is also toxic and leads to synaptic impairment and dopaminergic neural loss [65]. According to that observation, a theory of loss-of-function of the native protein has been proposed as the pathological mechanism driving PD. Loss-of-function of the soluble bioactive form of α-syn was also observed as a result of the aggregation process. Aside from *SNCA*, other disease-related mutations may impact the α-syn aggregation dynamics, which will be discussed in detail in further sections.

### 2.2. Other Genes Related to α-Synuclein Pathology

According to the literature data, variations in more than 20 genes predispose to development of monogenic familial PD as well as other α-synucleinopathies [66]. The products of these genes are implicated in critical cellular pathways, and their dysfunction has also been attributed to the sporadic forms of the disease. Mutations in *GBA*, *LRRK2* and *VPS35* are linked to dysfunction in the autophagy–lysosomal pathway (ALP) and they induce AD PD, whereas mutations in *PRKN*, *PINK1* and *DJ-1* are associated with mitochondrial pathways and cause autosomal recessive (AR) PD [66,67]. Overall, genetic variations increase the risk of developing PD up to 25% [68].

*GBA* gene mutations are the main causative agent of a rare lysosomal storage disorder, Gaucher disease, and the most common genetic risk factor for PD (7–10% of PD patients) [69]. *GBA* mutation carriers also have an increased risk of developing DLB, and in some cases MSA [70]. *GBA* encodes for glucocerebrosidase (GCase), a lysosomal enzyme that breaks down glucocerebroside into glucose and ceramide. Decreased GCase activity may lead to the accumulation of proteins processed through the ALP, including α-syn, in neurons. Accumulation of GCase substrate glucocerebroside leads to alterations in glycosphingolipid homeostasis, which affects membrane composition, vesicular transport and in turn promotes α-syn aggregation. Conversely, accumulated α-syn decreases lysosomal activity via inhibition of trafficking of GCase (and other enzymes) from the ER to the lysosome, which renders a vicious cycle [71]. There is compelling evidence that *GBA* mutations are inevitably linked to ER stress. In vitro studies have demonstrated that retention of mutant GCase in the ER directly triggers ER stress and activates the Unfolded Protein Response (UPR) [72]. Misprocessing of mutant N370S GCase in the ER of dopaminergic neurons resulted in ER stress and increased release and propagation of α-syn [73]. On the other hand, α-syn overexpression was found to promote misfolding and aggregation of GCase by inducing ER dysfunction and inability to activate UPR [74]. It has been found that sporadic PD patients have significantly decreased activity of GCase in the affected brain regions [75], which could be either a cause or a result of α-syn accumulation.

*LRRK2* gene encodes for the leucine-rich repeat kinase 2 (LRRK2), a member of the Ras-of-complex (ROC) protein family, which possesses kinase and GTPase domains [76]. *LRRK2* mutations are present in 2–7% of all PD cases, and they are associated with late onset of the disease [77]. LRRK2 is involved in autophagy, vesicle trafficking and regulation of retromer complex [76]. Usually, mutations in *LRRK2* result in increased kinase activity, impairment of GTPase function or protein dimerization. Gain-of-function mutations of *LRRK2* contribute to aggregation of α-syn, exocytosis of toxic species and their propagation to other cells [77]. As in the case of *GBA*, numerous mutations and variants of *LRRK2* differently affect the levels of risk for developing PD [78]. Interestingly, a recent study has found a negative correlation between GCase and LRRK2 expression levels in PD patients, with a Rab10 being identified as a key mediator of the interaction [79]. LRRK2 is involved in upregulating key UPR chaperones, such as binding immunoglobulin protein (BiP), which promotes activation of protective pathways and cell survival. However, it was found that DA neurons carrying G2019S *LRRK2* mutation are more susceptible to ER stress and the resulting neurodegeneration [80]. In astrocytes, this pathogenic mutation exacerbates ER stress, impairs calcium homeostasis, disrupts mitochondrial dynamics and ultimately triggers the pro-apoptotic branch of the UPR [81,82].

The other essential genes predisposing to PD have recently been reviewed [83]. It is also highly likely that there are additional yet undiscovered genetic factors that may contribute to PD acquisition, and this should be addressed by further research studies.

### 2.3. Post-Translational Modifications

Considering the intrinsically disordered nature of α-syn monomer, multiple PTMs of the protein have been identified. Of note, a number of PTMs are a characteristic feature of α-syn deposits present in LBs [84].

Phosphorylation of α-syn at S129 is one of the most recognizable PTMs, with pS129-α-syn being found in LBs in PD and DLB patients’ brains (up to ~90% of total α-syn) [85]. Whether this modification in fact enhances the aggregation process or, conversely, is neuroprotective still remains a matter of debate. A number of distinct proteins, including polo-like kinase 2 (PLK2), casein kinase, G-protein-coupled receptor kinase 6 or phosphoprotein phosphatase 2 (PP2A), are implicated in regulation of α-syn phosphorylation/dephosphorylation at S129 [86]. However, it is not recommended to target this particular PTM until its actual significance to the pathology is well-defined. Other serine and tyrosine residues of α-syn, including S87, Y125, Y133 and Y136, have also been identified as phosphorylation sites [58]. Phosphorylation at S87 or Y125 is in general regarded as protective, but more data are needed to confirm this statement. Also, phosphorylation of these residues cannot be executed separately from others [84]. It is possible that phosphorylation at a specific residue may induce conformational shift and alter binding properties of α-syn, which affects aggregation. Further, the overall effect of phosphorylation might be dependent not only on the particular residue but also on the ratio of the phosphorylated/total amount of the protein, and the phosphorylation might be protective (or toxic) to a specific threshold.

The O-linked β-N-acetyl glucosamine modification (O-GlcNAcylation) is a specific PTM in which an uncharged acetylated hexosamine sugar is added to or removed from a serine/threonine residue of the target protein. The reaction is catalyzed by O-GlcNAc transferase or O-GlcNAcase, respectively [87]. At least nine O-GlcNAcylation sites within α-syn have been identified, and most of them involve serine and threonine residues in the NAC region [88]. Increasing levels of O-GlcNAc were found to positively correlate with the accumulation of monomeric α-syn [89]. On the other hand, O-GlcNAcylation at T72 or S87 protects α-syn from calpain-mediated cleavage, which would otherwise enhance aggregation of the protein [88]. A single O-GlcNAcylation at T72 blocks both oligomerization and fibrillization of α-syn, without affecting membrane-binding capabilities [90].

Truncation is an enzymatic cleavage that typically occurs in the C-terminal region of α-syn, around 121 residue [51]. C-end truncation constitutes a part of physiological α-syn processing, but truncated form of α-syn is found in higher amounts and within the aggregates in PD brains [60]. It is believed that this PTM enhances α-syn propensity to aggregate into toxic oligomers and fibrils [59]. Especially, calpain-mediated cleavage of α-syn is thought to contribute to PD pathology. Calpain is a calcium-dependent protease, predominantly expressed at presynaptic terminals and activated by increased calcium influx. Calpain has been shown to cleave monomeric α-syn after 57 residue and within the NAC region, or fibrillar α-syn around 120 residue [91]. Calpain expression is increased in PD brains, and its activity is linked to α-syn-induced toxicity, seeding and neurodegeneration [92]. Cleavage of soluble monomers by calpain inhibited fibrillization, whereas processing of the fibrillar α-syn promoted further co-aggregation of the full length protein as generated C-terminal fragments retained their fibrillar structure [93]. Interestingly, A53T mutant seems to be resistant to calpain-mediated cleavage [91]. The data obtained suggest that the cleavage site differs between various species of α-syn and that it determines the further α-syn propensity to aggregate. Caspase I, despite its pro-inflammatory and necrotic activity, was also shown to truncate α-syn and thus generate aggregation-prone species in vitro [94]. An enzyme implicated in α-syn clearance is a serine protease neurosin (or kallikrein-6), which is predominantly expressed in the CNS. Neurosin cleaves α-syn in the NAC region and thus enhances its degradation and inhibits fibril formation [95]. Additionally, cathepsin D and matrix metalloproteinases (MMPs) were also shown to cleave α-syn at C-terminus; more specifically, the presence of MMP-3 was noted in LBs [96].

As for other PTMs, ubiquitination is primarily regarded as a protective response that targets α-syn for degradation in the proteasomes. Several enzymes related to this process, including ubiquitin-specific protease 13 (USP13), may upregulate α-syn clearance [97]. However, upon exceeding the specific threshold as α-syn tends to accumulate over time, UPS becomes dysfunctional and no longer fulfills its tasks. In fact, α-syn in LBs can be found mainly as mono-, di- or tri-ubiquitinated [98]. On the other hand, nitration of the tyrosine residues of the protein (Y39, Y125, Y133 and Y136) has also been identified in the α-synucleinopathy brain specimens, but it exerted divergent effects on the protein aggregation and its actual role is not well-understood [99]. Acetylation primarily occurs at the lysine residues of the N-terminus (K6 and K10), and this modification increases α-syn membrane-bounding properties by stabilizing its α-helical structure. Acetylated α-syn is characterized by lower propensity to aggregate and different fibril structure. Correspondingly, sirtuin 2 (SIRT2) deacetylase has been shown to aggravate α-syn toxicity [100]. Lysine residues of α-syn can also be SUMOylated by the protein inhibitor of activated STAT2 (PIAS2). This modification prevents degradation of α-syn by disrupting the ligase-mediated ubiquitination process, but it has also been shown to inhibit aggregation and toxicity of α-syn in a separate study [101,102]. Such diverse effects could have resulted from the targeting of different residues by this modification. Oxidation of α-syn, which could be, for instance, driven by oxidized DA derivatives, decreases formation of fibrils and increases accumulation of toxic species (oligomers, protofibrils) [103]. An α-syn molecule can also undergo glycation by DJ-1, which presents both glyoxalase and deglycase activities. The glycation of the N-terminal lysine residues of α-syn by methylglyoxal (MGO) amplified the production of toxic oligomeric species [104].

Due to conflicting results obtained in separate studies, the roles of the respective PTMs of α-syn still remain unclear, and selecting the most suitable PTM to target in α-synucleinopathy remains a challenge. The potential utility of therapies based on modulation of α-syn PTMs will remain uncertain unless we fully understand the significance of the mentioned PTMs on the course and progression of the α-synucleinopathy.

### 2.4. Physiological Forms of α-Synuclein

α-syn, depending on its dynamic structure, may be located in different subcellular compartments, and distinct functions may be exerted. The protein can normally be found in the cytosol as a natively disordered soluble monomer or as α-helical multimers bound to negatively charged phospholipids of the cellular membranes and synaptic vesicles. These two forms of the protein (cytosolic and membrane-bound) are known to exist and interchange in an equilibrium [105]. It has been suggested that physiological α-syn may mainly present as α-helical tetramers, but this theory is controversial and has not yet been fully confirmed [106]. Under physiological conditions, α-syn is abundantly expressed in the presynaptic compartments, wherein it associates with vesicles and membranes and regulates vesicle trafficking, recycling, synaptic transmission and plasticity. Mechanistically, α-syn was shown to promote soluble N-ethylmaleimide-sensitive factor attachment protein receptor (SNARE) complex assembly via direct interaction with a SNARE component, synaptobrevin-2/vesicle-associated membrane protein 2 (VAMP2) [107]. The complex is directly involved in neurotransmitter release, including that of DA. It has been proposed that, after sustained periods of firing, α-syn reduces neurotransmission by inhibiting the refilling of the readily releasable vesicles [108]. Other synaptic proteins that are known to interact with α-syn are phospholipase D2, synapsin 3 and Rab small GTPases [43]. α-syn is also implicated in other essential processes, such as stabilization of mRNA in P-bodies, DNA repair, function of immune cells and DA biosynthesis [109,110,111,112]. It is thought that it is the α-helical structure and ability to bind membranes that provide α-syn the neuroprotective role as it was found that membrane-binding-deficient A30P mutant failed to show neuroprotection [113]. Nevertheless, a detailed physiological role of α-syn remains elusive and is to date not well-understood.

Due to the disordered nature and high flexibility of the protein, α-syn may swiftly adopt different conformations and is susceptible to generate distinct interactions with other molecules and to multimerize. Conformational shift occurs upon stress conditions or interaction with other proteins, lipids and specific ligands [114]. The particular types of conformational and oligomer states that α-syn can take in different cellular compartments, as well as specific determinants that regulate the conformational shift, remain unknown. Under pathological conditions, monomers of α-syn aggregate into higher-molecular-weight assemblies with β-sheet-rich structure [14]. Interestingly, in contrast to monomer, the α-helical tetramer of α-syn was shown to be resistant to fibrillization [105]. Regarding the essential role of α-syn in maintaining synaptic homeostasis, loss of function of the protein can lead to deleterious consequences. As α-syn monomers are displaced from their physiological location and trapped within amyloid fibrils, the α-syn pool in the synapse is depleted and neurotransmission is significantly impaired. Decreased levels of total α-syn were also detected in the cerebrospinal fluid (CSF) of patients with α-synucleinopathies [115]. This led to the conclusion that monomer supplementation should be considered as a form of therapy, but, on the other hand, supraphysiological levels of monomeric α-syn can also have detrimental consequences. α-syn overexpression was demonstrated to impair intersynaptic trafficking, decrease the reserve pool of vesicles and reduce neurotransmitter release, primarily due to disruption of the SNARE complex [116,117]. It also cannot be excluded that monomers may induce toxicity in yet unprecise mechanisms independent from aggregation, for instance, retention in cellular compartments or aberrant interactions with other molecules and signaling pathways. As exogenously derived α-syn may also build up into the existing aggregates, inhibition of the oligomerization and nucleation process would be a more appropriate approach. Altogether, it seems that the proper balance between various forms of α-syn (soluble monomers and insoluble aggregates) is crucial for proper functioning of neural cells.

### 2.5. α-Synuclein Oligomers

Unfolded monomers of α-syn multimerize following a specific pattern: first, they assemble into soluble oligomeric species, followed by elongation to protofibrils, to eventually form typical amyloid deposits. Oligomerization of native α-syn can be driven by multiple biological factors, including normal aging, genetic mutations and polymorphisms, PTMs, infections, exposure to toxins, metal ions and other pathological proteins, lipid imbalance or oxidative stress [18,19,20,77,118,119,120,121]. There are many different oligomeric strains, which are of variable sizes and shapes, and rich in β-sheets. Oligomers comprise several to dozens (usually 2–150 mers) of α-syn monomers that are connected by non-covalent bonds [122]. Like monomers, they are usually soluble and quite unstable due to dynamic changes in their conformation. Depending on the ability to form amyloid aggregates, they can be classified as on-pathway or off-pathway oligomers. Oligomers may take a form of ring, string, cylindrical, spherical or chain-like structure [123]. It is regarded that ring/annular oligomers are in general off-pathway, and thus they present toxic properties. Small spherical forms have also been shown to exert toxicity [124]. It has been proposed that the initial dimerization of α-syn at the membrane surface (which involves residues 96–102 at the C-end) is the key driver of oligomerization [125]. During the oligomerization process, the α-helical-rich intermediate species are formed first, and, as the transformation into fibrils progresses, the α-helical content decreases in favor of formation of β-sheet structure. Importantly, both α-helical and β-sheet-type intermediates may exert toxicity upon interaction with membranes and promote further aggregation [126]. In contrast to fibrils, which have parallel β-sheet structure, oligomers possess a rather anti-parallel structure, which facilitates their toxic properties, e.g., ability to create pores and disrupt membranes [127]. A low-resolution structure for on-pathway oligomers has been provided, which revealed a wreath-like shape, similar to that of other amyloid protofilaments [128]. A metastable oligomer of α-syn, which can be converted into either on- or off-pathway assemblies, has also been identified, and it consists of 11 monomers, is disordered and spherical in shape [129]. Nevertheless, the exact molecular structure of these and other species is yet to be determined. Given the heterogeneity and highly dynamic structure of oligomers, establishment of the specific pathological strain, which could be targeted by novel therapies, is difficult to achieve.

Oligomeric and protofibril species are believed to exert the greatest neurotoxic effect and seed further aggregation. Various α-syn oligomeric species may exert toxicity in neurons and glial cells in a number of mechanisms. As in the case of other aggregation-prone proteins, α-syn oligomers and protofibrils bind to lipid membranes of ER, mitochondria, lysosomes and vesicles and increase their permeability. This results in damage and leakage of the mentioned organelles, increased calcium influx, ion dysregulation and subsequently caspase-3-mediated cell death [25]. The presence of oligomeric species has been confirmed in PD brain specimens, and they are mostly located in axons and presynaptic ends [130]. Therein, they may disrupt microtubules and damage dendrites, as well as affect the axonal transport of synaptic proteins (e.g., synapsin 1) and synaptic vesicle docking, which impairs synaptic function [131]. Lastly, the oligomers are capable of inducing deleterious microglial activation [23].

It is not excluded that α-syn-induced neurodegeneration involves the multidirectional action of several different strains. The existence of various strains could provide an explanation for the different clinical and pathological presentation between individuals, either suffering from the same disease or from different α-synucleinopathies. It has been experimentally confirmed that different fibril strains of α-syn generate different oligomeric strains and have different levels of toxicity and propagation properties. Moreover, extracts obtained from PD, DLB or MSA patients’ brains, when inoculated into animal models, resulted in formation of the adequate pathology [14]. It is probable that the size and shape of the specific multimers influence their biophysical characteristics and thus make them more prone to affect the preferred type of cells, compartments or molecules. If this is the case, individualized therapies selectively targeting different strains of α-syn, and/or possessing specific cellular tropism, would be required for a single patient. However, to date, it is not known what the number of monomer units and the actual structure of the toxic strains are, and where the toxicity is derived from (PTMs, nucleation, fibrillation?). Detailed identification of the particular aggregated forms of α-syn that induce pathology, as well as determination of the most prevalent ones, remain an area of active investigation.

### 2.6. Fibrillar Strains of α-Synuclein

Formation of fibrils and higher-order assemblies of α-syn is a multistep process. The oligomerization is followed by primary nucleation, fibril growth/elongation, secondary nucleation, maturation of filaments, transmission, seeding and amorphous aggregation [132,133]. It is thought that the main culprit for development of α-synucleinopathy is not the presence of fibrils per se but rather the entire process of deposits formation [134]. When the natural equilibrium between α-syn oligomers and monomers becomes disrupted (with a higher ratio for oligomeric forms), oligomers undergo a slow conversion into fibrils. Long-term incubation also facilitates the aggregation process, which is the case either for wild-type or mutant α-syn [124]. It has not been determined so far as to whether the pathological conversion of α-syn is triggered directly, by the existing aggregates in a seeding process, or is an indirect response to cellular stress conditions.

Upon higher concentrations of α-syn, the protein–lipid ratio in the membranes increases, which facilitates the local nucleation of α-syn and amyloid formation [135]. The aggregation follows a typical nucleation–polymerization pattern, with three main phases: lag phase, elongation phase and plateau phase. Initially, the newly formed oligomers act as nuclei that seed further aggregation. Such initial seed nuclei, due to disrupted proteostasis, are able to escape cellular clearance mechanisms [136]. Subsequently, fibrils grow in size and number as a result of cooperative monomer addition, until they reach adequate length and maturity [124]. Other associated events of amyloid aggregation process include secondary nucleation (fibril fragmentation), transmission/exocytosis and seeding to other cells and brain regions. It has recently been found that the fibrillar surface serves as a catalytic region for secondary nucleation, and the two putative binding pockets implicated in this process have been identified, namely H50 and E57 [137]. The aggregation of α-syn can also be affected by several environmental factors, like acidic pH, presence of salt, lipid vesicles or preformed fibrils (PFFs), which altogether impact the intermolecular interactions of the protein [124,133,138]. Such amyloid aggregates formed under various solution conditions, despite sharing a common cross-β fold, are characterized by distinct physicochemical properties. Different properties of the fibrillar strains may determine the particular lesion profile and dissemination in the specific brain region. In the case of MSA, distinct strain conformation (fibril or ribbon) determined the disease severity and progression in vivo [139]. Recent studies based on Seed Amplification Assays (SAA) have demonstrated that the type of detected α-syn strains may allow to distinguish between different α-synucleinopathies [140]. In PD and MSA samples, α-syn strains significantly differed in terms of conformational properties and proteinase K resistance profile. Strains found in MSA exhibited higher proportion of β-sheets, faster aggregation kinetics and considerably higher toxicity [141]. A separate study has demonstrated that DLB samples were characterized by faster seeding and higher fluorescence intensity than those obtained from PD patients [142]. It has also been reported that PD and DLB seeds assemble into paired or bundled multiple filaments, whereas MSA seeds tend to from longer, undivided twisted or straight filaments [143]. The amplification techniques are highly sensitive and specific in detecting and differentiating α-synucleinopathies and therefore constitute an area of extensive research.

The insoluble amyloid fibrils are in general characterized by a typical cross β-sheet structure, in which parallel and anti-parallel β-sheets run perpendicularly to the fibril axis [127]. It is believed that the cross-β cores of the large fibrillar deposits prevent the exposure of hydrophobic patches to damaging interactions. The increased propensity of α-syn to misfold and form β-sheet-rich amyloid assemblies is attributed to the NAC domain [7]. Fibrils eventually assemble into proteinaceous cytoplasmic inclusions, termed as LBs and Lewy neurites (LNs). These intracellular structures constitute a neuropathological hallmark of PD, DLB and other α-synucleinopathies. Aside from neurons, they can also be found in astroglial cells in DLB and MSA brains [9]. Lewy structures consist of a dense core of aggregated protein surrounded by loose filaments. They are composed of α-syn in distinct forms of various molecular weight (mainly misfolded, insoluble and fibrillar strains, S129-phosphorylated, C-truncated and ubiquitinated) as well as other proteins, membrane fragments and dysmorphic organelles [9,85,134]. The most compelling theory assumes that LBs constitute a form of protective aggresomes that aim to reduce the toxicity of α-syn oligomers by converting them to stable, less dynamic fibrils. It has also been proposed that it is the accumulation of α-syn in LNs, not LBs, that is responsible for driving synaptic and neuronal dysfunction [144]. It is yet to be established if Lewy pathology is actually neuroprotective and/or neurotoxic, why the nerve cell is not able to function with α-synuclein deposits inside and what is the mechanism in which α-synuclein ultimately kills the cell (Figure 1).

Various strains of α-syn differ significantly in their conformational properties, aggregation kinetics, toxicity and affinity for diverse brain regions and cellular compartments. This diversity contributes to the heterogeneity of α-synucleinopathies. Innovative approaches discussed later in the paper show differing affinities and potencies for specific α-syn regions (N-terminus, NAC, C-terminus) or target distinct α-syn aggregates (monomers, oligomers, fibrils or species modified by PTMs). Therefore, in-depth recognition of the protein structure and distinct characteristics of α-syn strains are crucial to understanding and advancing targeted and personalized therapeutic approaches.

## 3. Endoplasmic Reticulum Stress Induced by α-Synuclein

Due to high demand in mRNA and protein axonal transport and intense translation of synaptic proteins (like α-syn), maintenance of ER proteostasis is essential for proper functioning of particularly susceptible DA neurons. ER is an organelle responsible for the production and quality control of newly translated proteins. Overload of cargo in the ER machinery leads to protein misfolding and accumulation. This in turn impairs calcium homeostasis, dysregulates the axonal transport and triggers ER stress conditions [145]. Multiple lines of evidence suggest that ER stress triggered by α-syn is particularly implicated in the pathogenesis of PD. Aggregates of misfolded α-syn are known to associate with and penetrate cell membranes, including ER membrane and mitochondria-associated ER membrane (MAM) [146]. Accumulation of mutant or overexpressed α-syn within the ER lumen interferes with the protein folding process, which induces ER stress and activation of the UPR [147]. It has been found that α-syn may directly interact with BiP chaperones and thus induce UPR [148,149]. The UPR consists of three major stress sensors, IRE1α (inositol requiring enzyme 1α), PERK (PKR-like ER kinase) and ATF6 (activating transcription factor 6), among which PERK is regarded as critical for determining cell fate. PERK also presents some additional activities: it induces nuclear-factor-E2-related factor 2/antioxidant response element (Nrf2/ARE) pathway to protect cell against oxidative stress and regulates lipid metabolism via phosphoinositide 3-kinase (PI3K)/Akt signaling [150].

BiP chaperones sense misfolded proteins, including α-syn. Upon interaction with misfolded α-syn, BiP dissociates from IRE1α, PERK and ATF6, and the three UPR sensors are activated. Activated PERK phosphorylates the α subunit of the eukaryotic translation initiation factor 2α (eIF2α), which aims to restore proteostasis via transient inhibition of protein synthesis and selective upregulation of target genes, encoding for ER chaperones and ER-associated degradation (ERAD) components. If this approach fails, the cell is targeted to programmed cell death via activation of PERK-dependent, pro-apoptotic C/EBP-homologous protein (CHOP) [149]. Activation of the other UPR sensor, IRE1α, induces cytoprotective splicing of the *X-box binding protein 1 (XBP1)* mRNA, but it may also trigger c-Jun N-terminal kinase (JNK) phosphorylation and apoptosis upon chronic stress conditions [151]. ATF6, when activated, undergoes ER–Golgi transfer and subsequent cleavage to the cytoprotective, transcriptionally active form ATF6n [152]. All three UPR sensors cooperate in maintaining cellular proteostasis, but the specific interactions between them are complex and to date not well understood. Aside from UPR activation, α-syn accumulation induces morphological dysfunction of the ER, disrupts ER chaperone levels and affects Golgi–ER protein transport and ER–mitochondria calcium transfer, which impairs proper ER functioning [153]. Prolonged ER stress leads to mitochondrial damage and cell death in a positive feedback loop, via generation of reactive oxygen species (ROS) and calcium leakage, which further potentiates α-syn accumulation and cellular impairment [19]. Importantly, new findings have demonstrated that extracellular α-syn PFFs can induce ER stress and alterations in iron metabolism by affecting expression of crucial iron-homeostasis-related proteins: divalent metal transporter 1 (DMT1), ferroportin (FPN), iron regulatory protein 1 (IRP1) and hepcidin. The disturbances in iron metabolism were apparently mediated by ER stress as they were similar to those induced by the ER stressor, thapsigargin, and could be blocked by treatment with the ER stress inhibitor, salubrinal [154]. The above-mentioned factors altogether contribute to neurodegeneration in α-synucleinopathy.

There is increasing evidence that UPR may be the main pathway involved in α-syn-related cytotoxicity as targeting of UPR either by genetic approaches or pharmacological modulators proved effective in preclinical models of PD. Increased levels of the relevant UPR/ER stress markers, like BiP, p-PERK and p-eIF2α, have been found in brain specimens from PD patients [75,155]. Interestingly, despite the common belief that chronic activation of PERK is the main culprit for neurodegeneration, in vivo studies have demonstrated that genetic depletion of PERK alters axonal DA release and uptake, which leads to DA neuron loss and PD-like motor dysfunction [156,157]. Therefore, it seems that proper balance of PERK activation is essential for maintenance of proteostasis and neuroprotection [158]. Overexpression of α-syn has been shown to induce DA neuron apoptosis via the PERK/CHOP-dependent pathway, and this effect was compromised by gene delivery of BiP [149]. The other study has suggested that α-syn may induce neurodegeneration in both a CHOP- and caspase-12-dependent manner [159]. In astrocytes, mutant α-syn was demonstrated to activate CHOP, induce Golgi fragmentation, promote calcium influx, decrease the level of glial-cell-derived neurotrophic factor (GDNF) and neurite outgrowth [24,160]. Apart from PERK, α-syn oligomers were shown to preferentially induce splicing of XBP1 [161,162] as well as JNK-dependent cell death [151] in preclinical models of PD. Conversely, genetic ablation of *XBP1* was suggested to trigger ER dysfunction, α-syn accumulation and neurodegeneration [163]. α-syn also interacts with a protective factor of the UPR, ATF6, and inhibits its ER–Golgi transit in COPII vesicles. This prevents ER-stress-mediated activation and processing of ATF6 and enhances pro-apoptotic signaling [152]. Oligomers can also disrupt the VAMP-associated protein B and C-protein tyrosine phosphatase interacting protein 51 (VAPB-PTPIP51) pathway via binding VAPB, which weakens ER–mitochondria association [146]. Moreover, a number of PD-related mutations (e.g., *GBA*, *SNCA*, *PRKN*, *ATP13A2*) are somewhat associated with ER stress activation and dysregulation of UPR mediators [75,153,164,165]. *PINK1* and *LRRK2* mutations result in MAMs dysfunction, and *LRRK2* was also shown to inhibit upregulation of BiP and PERK-mediated activation of E3 ubiquitin ligases [80,166]. Also, a much less common *PARK20* mutation is associated with large amounts of accumulated proteins, severe ER dysfunction and UPR activation in fibroblasts [167]. Overexpression of *DJ-1* induces cell death via activation of ATF4 [168]. ATF4 interacts with *PRKN* promoter [169] and spliced XBP1 (sXBP1) with *DJ-1* promoter to increase their expression and exert cytoprotective responses [170]. sXBP1 upregulates the expression of brain-derived neurotrophic factor (BDNF), which improves cognitive function and synaptic plasticity [171]. Other neurotrophic factors, like mesencephalic-astrocyte-derived neurotrophic factor (MANF) or cerebral dopamine neurotrophic factor (CDNF), were also shown to reside in the ER lumen and attenuate detrimental ER stress response. The mechanism by which MANF exerts neuroprotection is direct binding to IRE1α [172].

As a vicious cycle occurs between α-syn and ER stress, it is hard to determine which comes first, α-syn pathology or ER dysfunction. UPR may act as a double-edged sword depending on the context, and it remains unclear what triggers the switch between pro-survival and pro-apoptotic response. Thus, adequate modulation of UPR activity may be challenging. Notwithstanding this, alleviating ER stress and reestablishing ER proteostasis may represent a promising disease-modifying strategy. It also needs clarification if ER stress is directly triggered by mutant misfolded α-syn translated in ER lumen or indirectly by α-syn translocated from cytosol, and this pathogenic mechanism could also be considered as a potential therapeutic target.

Apart from ER stress, α-syn is implicated in a myriad of other intracellular and extracellular events. The molecular perturbances associated with α-syn pathology have recently been reviewed [173], and the most important aspects of α-syn toxicity are schematically presented in Figure 2.

## 4. Drugs Targeting α-Synuclein

### 4.1. Gene Therapies

Gene therapies hold great promise for the potential treatment of genetic forms of α-synucleinopathies as they may emerge as first-line, casual treatment strategies. Multiple attempts regarding substitution of defective genes (via vector-mediated delivery or small-molecule modulators) have been undertaken. Silencing of other genes (like *SNCA* or *LRRK2*) can be achieved by RNA interference (RNAi) technologies (small interfering RNAs/siRNAs and short hairpin RNAs/shRNA), antisense miRNAs or oligonucleotides (ASOs). Less common strategies involve zinc finger nucleases, ribozymes, targeting transcription factors, epigenetic modifications or repressing of gene promoter. A quite recent CRISPR/Cas9-based technology was also suggested to correct PD-associated genetic mutations [174]. A majority of concepts aim to target PD-related genes or to transcriptionally regulate *SNCA* expression. Alternatively, the level of protective proteins, like growth factors, artemin, neurturin, persephin or Nurr1, can also be adjusted by therapeutic overexpression [175,176].

Most of nucleic-acid-based strategies require delivery by liposomal or viral vectors to achieve satisfactory bioavailability in the CNS [35]. Nucleic-acid-based therapeutics delivered by viral vectors were effective in vitro and in vivo. ASOs are regarded as supreme to siRNA technologies, and the first ASO-based therapy has been approved by the Food and Drug Administration (FDA) for the treatment of MSA and other neurological disorders. ASOs could be useful in terms of correction of splice-site mutations and splicing defects (as in the case of *DJ-1*) [177]. Adeno-associated virus (AAV) serotypes 2 and 9 demonstrated an excellent tropism for the CNS, ability to cross the BBB and, in contrast to lentiviral vectors (LVs), they have limited risk of insertional mutagenesis upon integration into the host genome. Delivery of the key enzymes of DA biosynthesis, such as tyrosine hydroxylase (TH), aromatic L-amino acid decarboxylase (AADC), glutamic acid decarboxylase (GAD) and GTP-cyclohydrolase (GCH1), by AAVs has also been reported, but this approach seems to be rather symptomatic [178]. Recently, it has been suggested that *SNCA* transcription can be regulated by tripartite motif (TRIM) proteins, TRIM17 and TRIM41, which may be of relevance for new gene therapies [179]. DNA methylation at *SNCA* intron 1 was also effectively targeted by CRISPR/Cas9-based system, and the therapy reduced *SNCA* expression [40].

Despite preclinical effectiveness, there is a major concern in delivery of gene therapies in human patients. In many cases, this requires a repeatable invasive stereotaxic or intrathecal injection, which might not be accepted by some of patients. Potential delivery of these therapies at the time of other PD-related surgical procedures, like DBS, also generates some technical and practical issues. Administration of vectors directly into SN is unlikely to prevent α-syn pathology in the other neuroanatomical areas. Although novel viral vectors are designed to pass through the liver and spleen and cross the BBB due to their affinity to bind specific receptors in the BBB (e.g., ApoB [180]), use of viral vectors may lead to unpredictable effects, and the high cerebral bioavailability observed in animals is unlikely to be achieved in humans. Other proposed delivery methods for RNAi include use of rabies virus glycoprotein (RVG)-modified nanocarriers or liposomes [181], but these, on the other hand, may trigger adverse immune reactions. Moreover, intravenous administration is still not convenient in terms of long-term therapy and decreases compliance as it requires regular admission to healthcare points. Short peptides were also shown to effectively transport RNAi through the BBB [182], but these molecules are unstable and can easily be degraded by serum proteases. As an alternative, intranasal administration was recently proposed for delivery of RNAi and ASOs [183]. This approach is non-invasive and characterized by excellent BBB penetration, bioavailability, pharmacokinetics and pharmacodynamics, although it allows to administer only a limited volume of the drug. Systemic administration, on the one hand, renders greater access to periphery than CNS, but, on the other hand, some aspects of α-synucleinopathy (e.g., autonomic dysfunction) are particularly attributed to the PNS. Also, a study on novel compound ENT-01/Enterin has proven that PD may indeed be treated through the gut [184].

Another important negative aspect of gene therapies is that α-synuclein is expressed not only in the CNS but also in other tissues and organs in the periphery, especially in red blood cells [185]. The physiological function of α-syn in the periphery is largely unknown. Also, given the essential role of α-syn at endogenous levels in the CNS, complete silencing of its expression could be detrimental. Studies confirmed that nigrostriatal degeneration directly results from *SNCA* knockdown [186]. Therefore, manipulation in the expression level of α-syn should be conducted carefully, to a certain extent, in order not to lead to serious implications due to excessive reduction in physiological α-syn. The therapeutic window for α-syn knockdown is yet to be established, but it is speculated that levels greater than 90% could be deleterious [187]. It is also to be clarified whether *SNCA*-targeted approaches would be suitable for sporadic patients or those bearing other mutations. If these limitations were overcome, gene therapies would be an excellent choice for genetically affected patients with inherited PD.

Small molecules represent a great alternative to classical genetic approaches. Recently, synucleozid was shown to effectively inhibit translation of α-syn mRNA in vitro [188]. Posiphen and phenserine decreased *SNCA* expression in preclinical models, but the latter is regarded as non-specific [189]. Several FDA-approved β2-adrenergic receptor agonists (clenbuterol, metaproterenol, salbutamol) were found to reduce *SNCA* expression in high-throughput screening (HTS). Mechanistically, the effect was based on decreasing histone 3 lysine 27 (H3K27) acetylation of the promoter and enhancer regions of *SNCA*. The compounds passed the BBB and effectively reduced α-syn levels in a dose-dependent manner, both in vitro and in vivo. Conversely, β-blocker propranolol was found to increase the levels of *SNCA* mRNA and, as a result, α-syn levels [190]. Some studies indicate that salbutamol may decrease risk for developing PD; however, this remains controversial as it could be rather related to a specific cohort and cigarette smoking [191]. Also, as salbutamol acts on β2-adrenergic receptors, it is non-specific for α-syn pathology and may induce cardiovascular side effects.

### 4.2. Natural Compounds

Not long ago, it was discovered that endogenous catecholamines (L-DOPA, DA, norepinephrine or epinephrine) may interfere with α-syn aggregation, via both covalent and non-covalent interactions with specific residues of the protein [192]. This, however, did not prevent α-syn toxicity but, conversely, aggravated it due to formation of new toxic species [193]. DA-derived quinones act as pro-oxidants and may inhibit a protective process of α-syn fibrillization [194]. Despite limited utility of these compounds, their structure may provide scaffolds for development of new, more suitable drugs. A large variety of exogenous natural compounds have been tested in preclinical models of PD, which include polyphenols (curcumin, resveratrol, baicalein, gallic acid/GA, ellagic acid/EA, nordihydroguaiaretic acid/NDGA, epigallocatechin-3-gallate/EGCG, rosmarinic acid, kaempferol, quercetin, resveratrol), alkaloids (berberine, isorhynchophylline) and terpenoids (celastrol, ginsenosides, squalamine and trodusquemine). They decrease α-syn toxicity either via direct interaction or indirectly, via activation of neuroprotective molecules and pathways. In general, almost all these compounds are characterized by a multifactorial mode of action, which involves a combination of anti-aggregative, anti-oxidative, anti-inflammatory and anti-apoptotic activities [195]. These molecules are known to alleviate ER stress in neurons by decreasing α-syn burden and switching UPR activation towards cytoprotective response, which mainly involves upregulation of the Nrf2 pathway [196,197]. The anti-aggregation effect of these compounds is induced by (i) decrease in α-syn expression (resveratrol), (ii) stabilization of α-syn molecules (baicalein, GA, NDGA, EGCG, ginsenoside Rb1), (iii) inhibition of oligomerization (curcumin, baicalein, EA), (iv) disruption of lipid–protein interaction (EA, EGCG, squalamine, trodusquemine), (v) destabilization of assemblies (curcumin, baicalein, EA, GA, EGCG, squalamine, trodusquemine), (vi) inhibition of fibrillization (curcumin, baicalein, cuminaldehyde, delphinidin, GA and EGCG), (vii) formation of non-toxic off-pathway aggregates (ferulic acid, EGCG), (viii) increase in α-syn clearance (celastrol, resveratrol, genipin, isorhynchophylline) [198,199,200,201,202,203,204,205,206,207,208,209,210,211,212,213,214,215,216]. Apart from α-syn, curcumin was shown to inhibit Aβ aggregation, and EGCG was reported to disrupt aggregation of a least 14 other amyloidogenic proteins [217,218]; these findings are important in the view of cross-seeding effect. What is more, some of these compounds exert unconventional, interesting activities: for instance, berberine accelerates production of L-DOPA by gut bacteria, genistein acts as a natural MAO inhibitor and trodusquemine stimulates the regeneration of injured tissues via recruitment of stem cells [219,220,221]. Some of the natural compounds and herbal extracts are currently undergoing phase II clinical trials for PD, which include DA 9805 (NCT03189563), hypoestoxide (NCT04858074) and WIN-1001X (NCT04220762).

Polyphenols seem to be the most promising class of molecules, with their properties being widely tested in vitro and in vivo in terms of neurodegeneration, cancer and cardiovascular diseases. They can usually be found in fruit, vegetables, herbal beverages, tea and wine. A common structural scaffold of these molecules is based on several phenolic moieties with one to three hydroxyl groups, and they may contain other functional groups and structural ramifications. The number, position at the phenyl ring and conjugation of the hydroxyl groups is critical for inhibitory activity towards α-syn [222]. Mechanism of action of polyphenols involves interaction with the charged and disordered C-terminus of α-syn and/or with the aggregation-prone NAC region. Polyphenolic compounds may form a Schiff base and bind α-syn aggregates through hydrophobic and hydrogen bonding, which destabilizes the β-sheet network of aggregates and promotes formation of nontoxic off-pathway fibrils [200,209]. Among phenolic derivatives, curcumin is of particular interest due to multifaceted action on α-syn confirmed by a series of studies, although there is poor evidence for the actual target engagement. Moreover, polyphenols are generally non-specific and may affect other off-target pathways and receptors. Their mechanism of action is not fully understood, and some potential adverse events may be difficult to predict. Also, due to poor solubility, stability and bioavailability, they would usually require very high dosages multiple times a day to achieve satisfactory pharmacokinetic and pharmacodynamic parameters. Improvement in these parameters can be achieved by development of analogic compounds or specific vectors (e.g., nanoparticles) that would allow for carrying the compound to the CNS. Some therapies based on synthetic curcumin derivatives or nanoformulations reduced α-syn aggregation in vitro [223,224,225]. These solutions are of greater economic impact than the original compounds as they can be commercialized. Nevertheless, polyphenols provide a major source of traditional medication, and their structural features may inspire development of new drugs by rational design.

There are several other classes of molecules that can potentially inhibit α-syn filament assembly. These include some antibiotics (rifampicin, geldanamycin, tanespimycin, polyene macrolides), dyes (Congo red, phenothiazines/methylene blue) and porphyrins [226,227,228]. Their precise mode of action is yet to be identified in preclinical experiments.

### 4.3. Monomer Stabilizing Agents

As growth and disassembly of α-syn fibers depend on monomer addition or dissociation, respectively, stabilization of monomeric forms emerges as a potential therapeutic strategy. One limitation is that the precise structure and position of hydrophobic pockets in α-syn monomer remain undefined. In spite of that, several agents have been developed and proven to stabilize monomers in experimental conditions, which include molecular tweezers and small molecules.

Molecular tweezers bind to positively charged aa residues of amyloid proteins, which disrupts hydrophobic and electrostatic interactions essential for self-assembly [229]. CLR-01 is one such compound, and it was shown to reduce α-syn toxicity across multiple studies. CLR-01 possesses a unique curved structure with a negatively charged concave cavity. The compound disturbs hydrophobic and electrostatic interactions by non-covalent binding to K10 and K12 residues at the N-terminus of α-syn; in this way, nucleation and aggregation are affected [230]. It can bind both monomeric and oligomeric species, is effective against both endo- and exogenous α-syn as well as several other amyloidogenic proteins [231,232]. CLR-01 is also capable of disassembling the existing fibers and stimulating formation of off-pathway species [233]. In contrast, a shorter analog of CLR-01, CLR-03, does not present inhibitory activity, probably due to lower number of aromatic scaffolds [231]. CLR-01 is non-toxic, crosses the BBB and shows favorable pharmacokinetics, but it did not manage to decrease the levels of aggregated α-syn in vivo and has not yet been considered for clinical trials [234].

Fasudil is a potent Rho kinase inhibitor, approved for the treatment of glaucoma in Japan and cerebral vasospasm in China. The compound was shown to cross the BBB and interact with the aromatic side chains of Y133 and Y136 residues at the C-terminus of α-syn monomer. This prevented nucleation, elongation and accumulation of α-syn in vivo, and improved motor and cognitive performance in transgenic mice [235]. Endosulfine-alpha (ENSA) is a low-molecular-weight protein that belongs to the cAMP-regulated phosphoprotein family. It has capacity to specifically interact with the C-terminus of membrane-bound α-syn, which prevents membrane-induced aggregation in vitro [79]. Short nucleic acids, which recognize specific protein targets, are termed aptamers. Further, 58-base DNA aptamers selected in a screening were found to bind α-syn with a high affinity, inhibit oligomerization and enhance degradation [236]. However, further mechanistic data on this effect of the aptamers are required.

### 4.4. Increasing the Clearence of α-Synuclein

Increase in α-syn intracellular clearance can be achieved by binding to chaperone-like molecules, with the key players being heat shock proteins (Hsps). These molecules assist refolding of nascent polypeptide chains, which prevents misfolding and aggregation events [237]. It is unknown how α-syn escapes chaperone-mediated protein quality control systems in the process of amyloid formation. It has been proposed that this could be due to very low kinetics of aggregation, Hsps becoming trapped within aggregates or too low expression level that cannot compromise overexpressed α-syn [238]. Multiple Hsp members, like CHIP (carboxyl terminus of Hsp70), Hsp40, 70, 90 and 110, are present in LBs [239,240,241,242]. All chaperones were shown to prevent α-syn aggregation, whereas heat shock transcription factor 1 (HSF-1) and Hsp90 modulate Hsp70 activity. Hsp70 is the best-studied and most promising chaperone, which has been shown to inhibit α-syn assembly via interaction of substrate binding domain with NAC or C-terminal region of prefibrillar α-syn [243,244]. As Hsp90 and Hsp70 exert opposing effects on target protein, most therapeutic strategies depend on Hsp90 inhibition or Hsp70 activation. Of note, chaperones are regarded as mainly cytoprotective, and such therapies are expected to have negligible side effects. Delivery of chaperones is difficult, however, and their expression can be modulated by gene therapy, natural compounds and synthetic molecules. Synthetic small-molecule inhibitory SNX compounds targeting Hsp90 displayed good pharmacokinetics and reduced α-syn oligomers in vitro, but they failed to protect against neurodegeneration in vivo [245,246]. Conversely, development of specific activators for Hsp70 or Hsp110 seems to be a promising approach, although pharmacological activation is harder to achieve than inhibition. It also requires validation as to whether the protective effect of chaperones would be equally effective at later stages of the disease.

GCase is essential for α-syn clearance and proper function of ER and lysosomes. GCase defect in PD can be addressed by a number of distinct approaches: gene therapy (PR001), small molecules (ambroxol, LTI-291, S-181, isofagomine, NCGC compounds), substrate reduction (venglustat, GZ667161) or receptor agonists (ESB-1609, fingolimod). The therapies are based on the substitution or replacement of the activity of dysfunctional GCase enzyme. Classical enzyme-replacement therapies (ERT) with recombinant glucocerebrosidase in Gaucher disease do not cross the BBB, nor do they affect neurological symptoms; thus, for PD treatment, these would require vector-mediated delivery (e.g., AAV) [78]. Administration of PR001A (AAV9-GBA1) is currently undergoing a phase I/IIa clinical trial (NCT04127578). Pros and cons of gene therapy have been described above. Small-molecule chaperones seem to be a promising alternative as they can bind to mutant GCase in the ER and enhance trafficking of the enzyme to the lysosomes. Ambroxol is an FDA-approved mucolytic agent, which restored lysosomal function and α-syn levels in *GBA* mutants in vitro and in vivo [247,248]. In a phase II study (NCT02941822), ambroxol passed through the BBB and increased the GCase activity in PD patients with and without *GBA1* mutation [249]. However, ambroxol administration is not recommended in patients with co-existing peptic ulcer disease and asthma. Other chaperone compounds, NCGC758 and NCGC607, effectively decreased levels of GCase substrates and α-syn in vitro [250,251], whereas S-181 exerted a similar effect in vivo [252]. LTI-291, a small-molecule allosteric modulator of GCase, presented a good dose-dependent BBB penetration and safety profile in a phase I trial [253]. The iminosugar isofagomine induced hyperacetylation of Hsp90, which enabled appropriate folding of GCase and increased its activity in vitro and in vivo [254,255]. Venglustat (other names: ibiglustat, GZ/SAR402671) is a brain-penetrant glucosylceramide synthase inhibitor that reduces the production of GCase substrates, glycosphingolipids. Despite good safety and tolerability in a phase I trial with reduction in glucosylceramide levels up to 75%, phase II study revealed that venglustat treatment deteriorates motor function in participants (NCT02906020) [256]. Having said that, a related small-molecule compound, GZ667161, presented activity towards mutant GCase with no adverse events in vivo [257]. The question is if these strategies are only restricted to *GBA* patients or if they could be applied in non-*GBA* patients as well given that GCase defect is also observed in sporadic cases. There are currently four ongoing trials on ambroxol effectiveness in PD and DLB patients without *GBA* mutations, which could provide us an answer for this issue (NCT05778617, NCT02941822, NCT04405596, NCT04588285).

Increased LRRK2 activity due to mutations may be compromised by gene therapies (miRNA-205; ASOs: BIIB094 and ION859) or small molecules (DNL151, DNL201, MLi2, PF-06447475, JH-II-127) [78]. MLi-2 and PF-06685360 are currently being tested in preclinical models [258], whereas DNL201 and DNL151 are undergoing clinical phase I–III trials in subgroups with and without *LRRK2* mutations (NCT03710707; NCT05348785; NCT05418673). ASO-based therapies, despite invasive administration, carry the advantage of selective CNS delivery. The safety and tolerability of BIIB094 in PD patients is to be determined in a phase I trial (NCT03976349). As *LRRK2* is robustly expressed in lungs, kidneys and several immune cells, systemic administration of LRRK2-targeting therapeutics may raise some efficacy and safety concerns. In fact, LRRK2 inhibitors were suggested to cause mild reversible changes in lung morphology [78], and, thus, they should be administered with caution to patients with the co-existing pulmonary diseases. On the other hand, partial downregulation of LRRK2 activity (~50%) may be sufficient and at the same time not induce adverse events in other tissues and organs.

### 4.5. α-Synuclein Aggregation Inhibitors

Degradation of α-syn oligomers and fibrillar forms seems to be one of the best possible options to target α-synucleinopathy as it can relieve overloaded UPS and ALP and restore neural function by direct removal of toxic deposits. To achieve this, the exact three-dimensional structure of pathogenic species and the mechanisms of oligomerization and toxicity need to be identified first. Alternative approaches involve stabilization or remodeling of oligomers and enhanced formation of non-toxic aggregates.

Catalytic drugs based on Co(III)cyclen have been proposed to selectively recognize and cleave target disease-related proteins. In vitro, the Co(III)cyclen containing peptide-cleaving catalysts destroyed pathogenic proteins related to Alzheimer’s disease, type 2 diabetes mellitus, Huntington’s disease and PD [259]. This effect is particularly desirable as pathogenic proteins are known to often co-aggregate and co-occur in one disease entity. However, the therapeutic potential of these catalytic drugs has not been assessed so far in in vivo conditions.

NPT100-18A is a cyclic, peptidomimetic compound, selected from a 34-compound chemical library, which targets the region of protein–protein interactions. This small molecule impairs formation of α-syn dimers on membrane surfaces via interaction with 96–102 residues at the C-terminus. This resulted in decreased amyloid formation and increased monomeric release in vitro. However, no conclusive results regarding NPT100-18A activity were obtained for the SN as for other brain regions [125]. Due to poor permeability of the BBB and brain bioavailability, an optimized version of the compound was developed under the name NPT200-11. However, significant pharmacokinetic optimization did not drastically improve target engagement or symptomatic relief as compared to the original compound [260]. Nevertheless, NPT200-11 has completed a phase I clinical trial (NCT02606682). A newly developed related compound, NPT520-34, has demonstrated BBB penetrance and good bioavailability profile. This molecule increased the levels of α-syn clearance via microtubule-associated protein 1 light chain 3 (LC3) upregulation, which led to reduction in proteinase K-resistant aggregates [261]. Another compound, NPT088, is a fusion protein combining immunoglobulin, phage-derived capsid protein g3p and a general amyloid interaction motif (GAIM). This motif allows for binding aggregates of multiple amyloid proteins. NPT088 reduced α-syn aggregates (including the proteinase K-resistant ones) and increased TH activity in mice overexpressing α-syn [262,263]. The molecule has advanced to phase I clinical trial for Alzheimer’s disease (NCT03008161).

Anle138b, a diphenyl-pyrazole (DPP) derivative, is a promising drug-candidate, discovered via HTS with medicinal chemistry optimization, which specifically targets and disrupts α-syn oligomers. Among a number of 10,000 compounds tested, anle138b appeared to be the most effective, of excellent oral bioavailability and BBB penetrance and innocuous to treated animals, even after high dose and long-term treatment. This small-molecule modulator inhibits the formation and accumulation of α-syn oligomers, whereas it does not interfere with the physiologic monomeric form of α-syn or alter the level of protein expression [264]. Anle138b was shown to bind a hydrophobic pocket of oligomers with high affinity (via interaction with I188, G68 and G86), which affects β-sheet formation [265]. Moreover, upon binding to α-syn aggregates, anle138b emits fluorescence so it can potentially be applied as a biomarker for detecting α-synucleinopathy [266]. Aside from α-syn, the molecule acts on a broad spectrum of other amyloidogenic proteins that often co-aggregate. Further, the compound was tested in multiple rodent models of prion disease, PD, MSA and Alzheimer’s disease with excellent results [267,268,269]. Not only did the compound provide significant neuroprotection by reducing protein accumulation but it also slowed the disease progression, even when applied as a post-treatment; it is so important because most cases of PD are diagnosed when the symptoms have already occurred. Although the compound showed effectiveness in several animal models, it failed to fully restore neural function in an acute neurotoxin-based MSA model [270]; therefore, combination with another compound, somewhat more specific to oxidative-stress-related apoptosis, should be considered. Also, in mouse models of genetic prion disease, anle138b appeared to be less effective as compared to sporadic model of this disorder [271]. The molecule has already completed two phase I trials in PD patients and healthy volunteers (NCT04685265; NCT04208152). One of these studies reported that the compound was well-tolerated, reached adequate plasma levels after intake of up to 300 mg and no severe side effects were observed [272].

SynuClean-D, similar to anle138b, was found to interact with aberrant aggregated forms of α-syn without affecting physiological monomers. The compound is effective against wild-type and mutant forms of the protein, and it was shown to impede both nucleation and elongation processes. SynuClean-D reduces the aggregation rate up to 70% in vitro, even at substoichiometric concentrations. The molecule is capable of dismantling already formed aggregates, with no apparent cytotoxicity of disrupted material, and it prevents seeding of α-syn. It also displays preferential activity towards specific species of α-syn, which suggests that its action is conformational-dependent [273]. Computational analysis revealed that SynuClean-D accommodates the inner cavity of α-syn fibrils in close proximity to the NAC region [274]. Two other anti-aggregational molecules, ZPD-2 and ZPDm, present structural and mechanistic similarities to SynuClean-D, with ZPD-2 targeting early and ZPDm targeting late stages of aggregation [275,276]. A further rational design approach based on SynuClean-D scaffold led to identification of MeSC-04. The molecule inhibits aggregation of α-syn via hydrogen bonds and van der Waals contacts with residues A53-V74 [277]. In view of the promising results, pharmacokinetic and pharmacodynamic properties of all the mentioned compounds should be further tested in animal models.

In a manner similar to the above-described agents, a novel compound, 03A10, was recently found to reduce the level of aggregated forms of α-syn in vivo, without interfering with the monomer [278]. This promising molecule warrants further testing in other disease models. In addition, an endogenous enzyme transglutaminase 2 (TG2) (catalyzing α-syn cross-linking and aggregation) was effectively inhibited by cystamine and cysteamine in vitro [279]. Other compounds reported to affect oligomerization include Bay K-8644, KYP2047, Demeclocycline HCl, Ro 90-7501 and SLS-007 peptides [280,281,282]. However, more experimental data are needed to assess their full inhibitory potential.

### 4.6. Endoplasmic Reticulum Stress Inhibition

Inhibition of ER stress can be executed by targeting UPR receptors, ER-specific enzymes, ERAD components or calcium homeostasis, with the first approach being the most popular. Interestingly, ER stress modulator salubrinal significantly alleviated PD symptoms in vivo via increase in eIF2α phosphorylation [283]. This suggests that inhibition of UPR downstream targets is not the only way to relieve ER stress, and that the outcome of UPR activity is not always clearly predictable. It has also been proposed that PERK inhibition does not always inhibit eIF2α phosphorylation as the latter is additionally regulated by several other kinases.

Commercially available GSK inhibitors of PERK (GSK2656157, GSK2606414) significantly decreased neurodegeneration in animal models of PD, prion disease and traumatic brain injury (TBI) [155,284,285]. Although effective, GSK2606414 exerted severe adverse events associated with pancreatic toxicity in tested animals. LDN-87357 is a highly specific PERK inhibitor selected in HTS, which is expected to cross the BBB. The compound rescued neurodegeneration in a cellular model of PD, and it could provide a promising alternative to GSK inhibitors [286]. It needs, however, in vivo testing to assess if pancreatic toxicity was particularly attributed to GSK or if this is a general side effect of PERK inhibitors. Importantly, there is growing evidence that either long-term inhibition of p-eIF2α or its chronic activation can lead to substantial cytotoxicity, and, therefore, partial inhibition or activation of eIF2α phosphorylation would be an optimal strategy, which is the case for LDN-87357. What is more, it is suspected that overly intensive disruption of protein aggregates or inhibition of oligomer transition to more stable, insoluble deposits could lead to ER stress and cell damage. Conversely, inhibition of apoptosis alone could be insufficient to rescue neural function in already dysfunctional neurons; this provides a rationale for combination therapy of UPR inhibitors with other anti-aggregative agents.

Integrated stress response inhibitor (ISRIB) is a small molecule that aims to derepress translational attenuation induced by eIF2α phosphorylation. ISRIB exerted a neuroprotective effect and had good tolerability profiles in prion-infected mice. In contrast to GSK inhibitors, ISRIB did not induce adverse effects on the pancreas [287]. It also significantly enhanced survival in a cellular model of amyotrophic lateral sclerosis (ALS) [158]. The other compounds that interfere with integrated stress response (ISR) are trazodone hydrochloride and dibenzoylmethane. Both compounds were found to abolish p-eIF2α-mediated translational attenuation. They were also apparently neuroprotective and non-toxic in mouse models of prion disease and frontotemporal dementia (FTD) [288]. Interestingly, sephin 1, which promotes eIF2α dephosphorylation, and guanabenz, which blocks eIF2α dephosphorylation, both provided neuroprotection in ALS models despite presenting opposite mechanisms of action [289,290]. It would be worthwhile to evaluate the efficacy of all the mentioned compounds in terms of α-synucleinopathy.

Despite strong evidence for the crucial role of UPR in PD, only several UPR modulators have been tested in α-synucleinopathy models so far. It needs clarification how targeting different UPR branches and/or different levels of the UPR cascade (ATF6, IRE1α/XBP1/JNK, PERK/eIF2α/ATF4/CHOP) affects cell survival. It is not known what the cellular implications are of the silencing of the respective levels and branches of UPR downstream pathways and which of them play a key role in PD pathogenesis. In contrast to PERK, which is abundantly expressed in the CNS, pancreas, bones, liver and kidneys, JNK3 is restricted to the CNS, and this could potentially be an interest of future research [291]. As UPR sensors are in general hyperactivated in PD, it is predicted that inhibition of UPR by specific modulators is unlikely to cause serious adverse effects in the peripheral organs and tissues.

### 4.7. Modulators of Mitochondrial Pathways, Oxidative Stress and Metal Metabolism

Direct targeting of mitochondrial dysfunction would be an interesting approach, given its pivotal role in PD pathology. Defects in *PINK1*, *Parkin* and *DJ-1* could be corrected by gene therapies. Therapies alleviating oxidative damage by means of ROS scavengers (coenzyme Q10, glutathione, vitamin B1, vitamin E) have also been proposed. Coenzyme Q10 was found to increase complex I activity and inhibit oligomerization of α-syn in a cellular model of PD [292]. However, the mentioned compounds are regarded as supplements rather than drugs, and there is no clinical evidence for their utility in monotherapy. Dual therapy of coenzyme Q10 with vitamin E was also not effective over a placebo in a phase III trial (NCT00740714). It is, however, possible to use these compounds in combination with other more specific drugs.

Therapeutical targeting of molecular interactions between α-syn and metals may be achieved by chelating agents, metalloproteases or metal-interfering molecules. These therapies are based on regulating the level of metal ion concentration or redistributing the metal pool in the affected brain regions.

Deferiprone is an iron chelator, approved by the FDA for thalassemia treatment. This molecule penetrates the brain and captures excessive intracellular iron to release it to the ECS or other iron-deficient cells. Several phase II trials assessed the effect of deferiprone treatment on PD course. In one study, deferiprone intake was associated with decreased iron deposit load in the SN and improved motor function in patients with PD, but the effect was more prominent at early stages of the disease (NCT00943748) [293]. However, in a separate study, no significant clinical improvement was achieved as regards both motor and non-motor symptoms in deferiprone-treated group (NCT01539837) [294]. Lastly, the newest trial has reported that PD patients who had never received L-DOPA experienced significant deterioration of symptoms after deferiprone treatment (NCT02655315) [295]. There are also some safety concerns regarding therapy with deferiprone as it may lead to agranulocytosis, neutropenia or hepatic fibrosis.

The other known iron chelator, deferoxamine, is used to treat acute iron poisoning. It has been proposed that intranasal administration of the compound could potentially be utilized in PD treatment [296]. D-607 is a molecule that, aside from iron-chelating properties, is known to exert a multidirectional effect on PD. The compound decreased α-syn aggregation, toxicity and activated DA receptors D2/D3 in animal models of PD [297]. PBT434 is a 8-hydroxyquinoline (8-HQ) derivative, which, apart from binding iron, was shown to decrease levels of oligomeric and fibrillar α-syn in murine models of MSA and PD [298,299,300]. The efficacy and safety of PBT434 were investigated in a phase I trial, in which the compound achieved a favorable and proportional pharmacokinetic profile, was non-toxic and well-tolerated [301]. Other chelators derived from 8-HQ include clioquinol, HLA20, M30, M32 and VK-28, among which M30 and HLA20 were also found to inhibit MAO-B [302,303,304]. An 8-HQ-derived compound and zinc ionophore, PBT2, was tested in several clinical trials for Alzheimer’s and Huntington’s disease with mixed results, but little is known about its effect on PD [305]. Above-described synucleozid is a compound that effectively decreases *SNCA* expression by targeting iron-responsive element (IRE) in the 5′-UTR of α-syn mRNA, and this approach exerted a protective effect in vitro [188]. In addition, metallothioneins are small proteins regulating copper/zinc homeostasis, which have also been proposed for potential neuroprotective PD treatment [306,307].

It is important to stress that metal metabolism is a significant aspect, but not the only one, in a complex pathology of PD. Therefore, it would be interesting to see the metal binding compounds in combination with other drugs relevant to PD. It should also be remembered that therapies based on metal chelating should not limit normal levels of these elements so as not to induce micronutrient deficiencies (e.g., iron deficiency anemia).

### 4.8. Agents Modifying Ubiquitin–Proteasome and Autophagy–Lysosomal System

Enhancement of autophagic processes could be of therapeutical benefit as it increases degradation and reestablishes physiological levels α-syn. Overexpression of ALP-associated proteins, like AMP-activated protein kinase (AMPK), cathepsin D, transcription factor EB (TFEB), lysosome-associated membrane protein 2A (LAMP-2A) or Beclin-1, provided neuroprotection in preclinical studies, but these findings need to be validated in complementary models [308,309,310,311,312]. The other proposed gene therapies are based on miRNA-7 and -124, which regulate expression of ALP-involved genes [174].

To enhance UPS, delivery of additional free ubiquitin molecules could facilitate labeling of α-syn and targeting it to degradation. This approach was investigated in fly model of PD [313]. Salidroside, a UPS modulator, was shown to enhance 20S proteasome activity associated with increased α-syn clearance [314]. Small-molecule IU1 improved proteasomal function and clearance of tau by inhibiting USP14 [315]. However, little is known about its effect on α-syn, and some reports indicate its potential toxicity [316]. A tetramethylpyrazine derivate, T-006, enhanced proteasomal α-syn clearance by activating both chymotrypsin-like UPS and mammalian target of rapamycin (mTOR)-dependent pathway [317]. Recently, specific nanobody constructs with a proteasome-targeting PEST motif have been developed (VH4PEST, NbSyn87PEST). These nanobodies could interact with NAC or C-terminal domain of α-syn, respectively, and target it for proteasomal degradation [318]. Nevertheless, all these therapies are still at the early stages of development and not much is known about their potential benefits.

As regards ALP, several inhibitors of the mTOR-dependent pathway were suggested to treat α-synucleinopathy, the most recognizable being rapamycin. Rapamycin treatment is associated with a multitude of limitations, like poor stability and solubility, lack of specificity, immunosuppression and other side effects (oral ulcers, impaired wound healing, hypertriglyceridemia, anemia, interstitial pneumonitis, secondary neoplasms); these effects are particularly prominent in long-term treatment such as that required in PD [319]. On the other hand, a novel glucan-particle-based vaccine combining α-syn antigen with rapamycin appeared to be more effective in vivo than humoral or cellular immunization alone [320]. MSDC-0160, apart from inhibiting mTOR, reduces pyruvate transport into mitochondria and rapidly improves mitochondrial oxygen consumption rate [321]. RTB101 is an inhibitor of mTORC1 (mTOR complex 1) that, by inducing autophagy, reduces the levels of GCase substrate glucosylceramide [78]. The drug was tested in a phase Ib/II trial in PD patients in combination with sirolimus, and the preliminary results provided by the development company seem promising. FDA-approved rapamycin derivative temsirolimus was also suggested to provide neuroprotection in parkinsonian mice [322]. However, the utility of all the mentioned drugs is limited as the mTOR pathway is involved in various cellular processes. There is a need for designing novel approaches, more specifically boosting ALP.

A disaccharide trehalose is one of the compounds that act via the mTOR-independent pathway. More specifically, trehalose increases lysosomal biogenesis and acts on several ALP-related proteins (Beclin-1, AMPK), which improves clearance of protein aggregates [323]. Interestingly, the compound was effective against PD upon oral administration in vivo, but not in vitro, upon exposure of cortical neurons to PFFs [324]. An FDA-approved compound 2-hydroxypropyl-β-cyclodextrin (HPβCD) increased MA-dependent α-syn degradation by activating TFEB [325]. LAMP-2A levels can be pharmacologically modulated by retinoic acid derivatives [326], but this approach may also affect cellular differentiation and metabolism. Approaches combining mTOR-dependent and -independent mechanisms have also been proposed, and a synergistic effect was observed upon treatment based on rapamycin and trehalose [327].

c-Abl is known to regulate the Akt/mTOR pathway, and enhanced activity of this kinase was detected in PD brains. Increased c-abl activity mediates phosphorylation at Y39 and aggregation of α-syn, as well as inhibits ubiquitination and activity of Parkin [328,329]. Nilotinib is a c-abl inhibitor approved for the treatment of chronic myelogenous leukemia and capable of crossing the BBB. In a phase II study in PD with dementia and DLB patients, nilotinib administration was safe, tolerable and reduced α-syn levels (NCT02954978) [330], although the validity of these data has been questioned [331]. Other known c-abl inhibitors currently tested in PD and/or DLB patients include K0706, FB-101, ikT-148009, radotinib, bosutinib and saracatinib (NCT03655236, NCT03996460, NCT04165837, NCT05424276, NCT04350177, NCT04691661, NCT03888222, NCT03661125). In general, c-abl inhibitors present unfavorable pharmacokinetics and small therapeutic windows, which limits chronic usage of these compounds.

### 4.9. Inhibition of α-Synuclein Propagation

Transmission-directed approaches involve removal of extracellular α-syn by a variety of clearance mechanisms, as well as decrease in α-syn release and uptake. These strategies also partially target endogenous α-syn as it is necessary for initiation of the spreading process.

Several endogenous proteases are known to degrade extracellular α-syn. Interestingly, the above-described Hsp70 chaperone was also reported to have that capacity [332]. Neurosin is a serine protease preferentially expressed in neural and glial cells in the CNS. The enzyme degrades α-syn and inhibits polymerization by cleaving the NAC region, but it is less effective against mutant forms [333]. It is speculated that neurosin acts by inducing proteolytic cascade of unidentified MMPs [334]. MMPs are involved in cleavage of extracellular α-syn, and, conversely, extracellular α-syn regulates MMPs’ activity [335]. On the other hand, MMP-3 mediated α-syn cleavage potentiates aggregation of the protein [336]. Plasmin is an extracellular serine protease particularly implicated in fibrinolysis. The enzyme is mainly synthesized in the liver, but it is also present in neurons and astrocytes in the CNS. In contrast to neurosin, plasmin cleaves both mutant and wild-type α-syn in a variety of forms (monomers, oligomers, fibrils) as well as several forms of Aβ [337,338]. By cleaving the N-terminus of α-syn, plasmin blocks spreading of released pathogenic protein and the resulting glial activation [337]. Neurosin was shown to colocalize in LBs, whereas plasmin activity was suggested to be dysregulated in PD [339,340]. However, these enzymes are not specific for α-syn and their overactivation could lead to degradation of other essential cellular proteins. For example, excessive plasmin levels may result in hyperfibrinolysis and augmented bleeding. Another limitation is that some of the α-syn strains are protease-resistant. Lastly, there are no pharmacological modulators of these enzymes or ERTs available yet, and their therapeutic effect was evaluated only via genetic overexpression.

An endocytic uptake of α-syn and other amyloid proteins is induced upon binding to heparan sulfate proteoglycans on the cell surface [341]. As α-syn enters the cell, it may escape endosomes in a manner similar to some viruses [342,343]. Although heparin and chloral hydrate inhibited heparan-sulfate-proteoglycans-mediated α-syn endocytosis in vitro [341], there is still a lack of more specific inhibitors of these macromolecules. Such a compound should delay α-syn propagation without affecting essential cellular processes.

As α-syn spreading is mediated by N-methyl-D-aspartate receptors (NMDAR), NMDAR antagonists like memantine have been proposed for PD treatment [344]. NMDAR inhibition could positively impact memory and learning processes, but excessive inhibition of the receptor may result in excitotoxicity. Inhibition of synaptic function could also aggravate the disease symptoms as the function of synaptic terminals is already weakened by the course of the disease [345]. Binding of extracellular α-syn to lymphocyte-activation gene 3 (LAG3) receptor mediates endocytosis of the pathological protein into neurons. LAG3 has been targeted by specific antibodies with good results [346], but the involvement of this immune receptor in disease progression is still under debate [347]. Toll-like receptors 1 and 2 (TLR1/2) have been targeted by small molecules and antibodies, and this approach alleviated neural immune response and neurotoxicity [348,349]. On the other hand, targeting of microglial or astroglial activity with non-steroidal anti-inflammatory drugs is unlikely to have any significant beneficial effect [350,351], and these drugs may increase risk of gastrointestinal, kidney and cardiovascular events. Also, it is likely that there are other yet-undiscovered proteins involved in extracellular clearance and propagation of α-syn.

### 4.10. Regulation of Post-Translational Modifications

Phosphorylation and dephosphorylation of α-syn are modulated by kinases and phosphatases, respectively. Among the kinases implicated in S129 phosphorylation, PLK2 is assumed to be the most selective for α-syn. BI2536 compound was shown to inhibit PLK2-mediated phosphorylation of α-syn [352]. On the other hand, modulation of the main pS129-α-syn phosphatase, PP2A, by coffee components (eicosanoyl-5-hydroxytryptamide and caffeine) mitigated the phenotype of parkinsonian mice [353]. Interestingly, the antidiabetic drug metformin also appeared to be an effective activator of PP2A, both in vitro and in vivo [354,355]. Still, the specificity of the mentioned enzymes is not particularly restricted to α-syn, and they are ubiquitously distributed in the organism. Modulation of their activity may impair crucial cellular functions, like cell division. It is also possible that some compensatory effects would compromise the therapeutic benefit [356,357]. It needs to be assessed which isoforms would be the most therapeutic, and also which approach is more beneficial to target, activation or inhibition. Considering all of the above, safe and efficient modulation of this PTM is particularly challenging.

As microglia-mediated nitration of α-syn leads to misfolding and toxicity, inhibition of inducible nitric oxide synthase (iNOS) was found to provide neuroprotection across several studies [358,359,360,361,362]. Cu(II)ATSM exerted a protective effect by preventing lipid peroxidation and α-syn nitration in vivo [363], and it has been investigated in a phase I trial with promising results (NCT03204929) [364]. A brain-permeable O-GlcNAcase inhibitor MK-8719 was also selected for phase I clinical trial [365], whereas thiamet G and ASN120290 O-GlcNAcase inhibitors were assessed in preclinical models [366,367]. Interestingly, MGO scavengers proposed for anti-diabetic treatment, namely aminoguanidine and tenisetam, mitigated glycation and improved clearance of α-syn in preclinical models [368]. A natural calpain inhibitor, calpastatin, as well as chemical inhibitors, Neurodur and Gabadur, reduced α-syn truncation in vivo [369,370]. Additionally, VX-765 improved neural cell survival in vitro by specifically inhibiting caspase I-mediated α-syn cleavage [94]. Much more data are needed to determine the specificity of these compounds, and, also, the exact role of these PTMs in α-synucleinopathy needs to be clarified.

### 4.11. Other Approaches

Some molecular events collateral to α-synucleinopathy could also be of therapeutic interest and their modulation potentially used in the form of adjunctive therapies. Given that α-syn pathology is associated with lipid dyshomeostasis, several lipid-associated agents have been developed. These include, among others, stearoyl-CoA desaturase (SCD) inhibitors 5b and YTX-7739, with the latter having successfully completed phase I study (NL9172) [371,372]. Oxidative stress related to α-syn can be managed by targeting NADPH oxidase, glyoxalase I or RAGE receptors [373]. Restoration in vesicular trafficking could be achieved by targeting Rab/SNARE proteins [374]. Regulation of sirtuin levels (SIRT1 overexpression or SIRT2 inhibition) could affect α-syn pathology at the epigenetic level [375]. Blocking of nuclear enzyme poly(adenosine 50-diphosphate–ribose) polymerase (PARP)-1 by veliparib, rucaparib or talazoparib also appeared to be protective against α-syn aggregation and phosphorylation at S129 [376]. Similarly, lovastatin, an agent approved for hyperlipidemia treatment, was found to attenuate α-syn aggregation and phosphorylation, as well as to enhance the tyrosine phosphatase 2 (SHP2)-mediated mitophagy [377,378]. ENT-01 induced α-syn displacement from the intracellular membrane-binding site in the enteric nervous system (ENS), and the compound restored bowel function and alleviated neuropsychiatric symptoms in a phase IIb study (NCT03781791) [184]. Neurotrophins (GDNF, MANF, CDNF) hold the potential of restoring degenerated neurons, but these proteins would require delivery by specific vectors or use of small-molecule mimetics. Moreover, CDNF was presented to reduce α-syn aggregation [379] and both MANF and CDNF to alleviate ER stress, neuroinflammation and apoptosis [380,381].

### 4.12. Passive and Active Immunization

The immunotherapeutic strategies against α-synucleinopathy can be generally classified as passive and active. Passive immunization is based on monoclonal antibodies against α-syn and vaccination on α-syn DNA or so-called affitopes (small peptides mimicking epitopes, inducing B-cell activation without deleterious T-cell response) [382]. Alternative immunization has also been proposed, and this utilizes vaccines with peptide-sensitized dendritic cells [383]. As a general trend, antibody-based therapies promote degradation of extracellular α-syn and inhibit propagation of the disease, but certain antibodies additionally induced autophagy-mediated clearance of intracellular α-syn [384,385]. To be effective, antibodies must be able to reach the CNS in large quantities and recognize α-syn in the cytoplasmic localization. Some technological advances with facilitated intracellular trafficking were also developed. Aptamers are short, single-stranded nucleic acids, which are non-toxic, not immunogenic and quite stable [236]. Intrabodies and nanobodies are small proteins that contain Fv variable region derived from antibodies and can act intracellularly. In this way, they are equally specific to antibodies but with improved brain penetrance and faster clearance [386]. Nonetheless, both these approaches require direct CNS delivery using viral vectors. As in the case of gene therapies, use of more invasive means of drug distribution is restricted to smaller treatment areas, and less invasive means are characterized by limited BBB permeability and potential systemic adverse effects. Most of the available antibodies bind all types of α-syn (monomers, oligomers, fibrils) at defined immunogenic epitopes (especially in the 89–140 aa region). This carries the risk of off-target reactions with other epitopes and substantial reduction in physiological α-syn level. In order to increase the precision of this strategy, an antibody should target a certain conformational epitope of α-syn and recognize only aggregated protein. It would also be desirable to design an antibody that would target several amyloid proteins simultaneously (e.g., a common NAC region) [387]. On the other hand, heterogeneity of strains may mitigate the immune response and drive resistance to antibodies. Further, therapeutic immunization may trigger nonselective autophagy and non-specific inflammatory responses [385,388]. The consequences of disassembly of extracellular α-syn with release of large amounts of soluble products are also unknown. The other challenge is to maintain high levels in the affected CNS region for prolonged periods, and this would require repetitive administration; this is especially the case for passive immunization, which does not induce immune memory. Despite these limitations, both passive and active immunization exerted neuroprotective effects in PD, DLB and MSA transgenic mice, and these approaches are now tested in clinical trials [384,389,390]. Detailed characterization of particular immunotherapies is beyond the scope of this review, but some of the most relevant strategies are listed in Table 1.

## 5. Summary and Perspective

Ample evidence supports the hypothesis that α-syn is the main culprit for PD and the related disorders, and targeting this protein could bring about the first disease-modifying approaches. Nonetheless, the disordered nature of α-syn, lack of structural information on its respective strains and complex aggregation mechanism all represent challenges in developing specific pharmacological modulators. The true pathogenic form, or rather forms, which elicit toxicity have not yet been identified. It is also not known whether cellular dysfunction and stress conditions are the trigger for α-syn misfolding, the result of α-syn aggregation or if they occur simultaneously in a vicious cycle. In view of the fact that a new report has indicated Epstein–Barr virus infection as a causative factor of another neurodegenerative disorder, multiple sclerosis (MS) [404], it cannot be excluded that α-syn prionic cascade may as well be originally initiated by a yet-unknown viral agent.

One major issue in PD treatment is that the disease is clearly heterogenous and multifactorial, and, hence, the pathogenesis may significantly vary between individuals. This may hinder the potential effectiveness of the applied therapies, and, therefore, it has been suggested that the therapeutic strategy should be specifically adjusted to the patient’s needs. This is, on the other hand, difficult to achieve, considering the fact that molecular changes are hard to detect in living individuals and there is still need for development of specific diagnostic tests and biomarkers for PD. Recently, milestone results were published from the Parkinson’s Progression Markers Initiative (PPMI) study led by the Michael J. Fox Foundation. A novel biomarker for α-syn was found, and it detects the protein in the CNS with high accuracy, at very early stages of the disease [405]. It seems that we are one step away from establishing optimized, less invasive screening programs that would allow for early identification and implementation of therapy in individuals at risk of PD. Moreover, a more personalized treatment approach, customized to patients’ individual genotype, may be achieved with increasing accessibility to genome sequencing. Considering the multitude of genetic mutations and SNPs implicated in PD pathology, routine genetic testing of individuals may be warranted to increase the therapeutic accuracy. Advances in biomarkers can also facilitate gathering of a more homogeneous patient cohort sharing a common molecular background, which can lead to more reliable results and greater treatment response in clinical trials. It has also been suggested that clinical trials in MSA are favorable for testing new drugs as the condition is fast-deteriorating and the duration of trial would be considerably shortened compared to the case of PD.

Basic research into disease pathogenesis is essential for identifying and testing new compounds, but it is not devoid of translational barriers. At present, no animal models are able to faithfully and fully reflect the human pathology, and this could be a reason for ineffectiveness of many therapies in clinical trials. Various animal species used in in vivo studies as well as different ways of inducing α-syn pathology (neurotoxins, viral vectors, PFFs, xenografts) may significantly vary between each other and thus render completely different results. Therefore, a potential drug should be validated in at least two models before clinical implementation. In vivo, vertebrate models should be preferable, in which α-synucleinopathy is induced by α-syn PFFs or AAV-mediated transgenic expression. Most studies were also focused only on pretreatment with the tested compound, which is in many cases clinically irrelevant. Additionally, preclinical studies on human samples that include post-mortem tissues, biopsies and biological fluids might be the most crucial to establish pathogenic mechanisms driving α-syn pathology.

A broad range of strategies against α-syn have been undertaken, most of them being at the preclinical or early clinical stage of development. Both targeting of α-syn directly (by gene silencing, small molecules or immunotherapy) and indirectly (by increasing clearance or reducing α-syn-related toxic events) are aimed at one common goal: to restore cellular proteostasis and halt neurodegeneration. A potential drug candidate must fulfill several critical criteria: be specific for α-syn, present no toxic and adverse events, have adequate biodistribution and evidence target engagement. In particular, the most promising strategy seems to be targeting of toxic oligomers by small-molecule chaperones or catalytic drugs. Small molecules can bypass the inability of protein-based drugs to cross the BBB and do not induce collateral immune reactions. These drugs, in most cases, dock into hydrophobic pockets or protein–protein binding sites of α-syn. Aside from blocking α-syn misfolding and aggregation, some of these compounds can also disentangle mature fibrils. In general, rationally designed small molecules are known to target mainly monomers and early aggregates, whereas compounds selected in HTS—late aggregates. As there are many toxic species that differ between individuals (or even in the same individual), as well as co-aggregates of other amyloid proteins, a perfect drug should be able to target many pathogenic strains at once. Also, the molecule must be specific for oligomeric α-syn and not disrupt monomers or other physiological proteins. Although it is doubted whether neurodegenerative brain changes can be reversed once initiated, some preclinical findings support this notion (including anle138b, induced pluripotent stem cells/iPSCs and growth factors). It is also difficult to classify clearly some of the proposed therapies or molecules as they present multiple modes of action at once. Such compounds with multifaceted activity can act on many pathological aspects of the disease, which could be of great benefit. Some potential off-target effects of the drug in other organs and systems may turn out to be an advantage in patients with multimorbidity. Recently, several GLP1 receptor agonists (semaglutide, liraglutide and lixisenatide) have demonstrated neuroprotective effects in preclinical models of PD and have now advanced to phase II trials [178]; therefore, these drugs could possibly be an excellent choice for diabetic patients with PD. Also, the above-described ambroxol and β2-agonists may be applied in PD patients with co-existing pulmonary disorders, which could also be regarded as a precision medicine approach.

Although non-invasive drug delivery through the BBB has always been regarded as problematic, some novel therapies have managed to bypass this limitation. Also, molecules targeting α-syn in the ENS appeared to be effective not only in prodromal stages but also in fully symptomatic individuals with neurological symptoms. A novel approach of magnetic-resonance-guided focused ultrasound provides the opportunity to reversibly open the BBB [406], but full safety and efficacy of this method are yet to be validated. There are also some advances in iPSCs that can be grafted into SN and there develop into DA neurons [407,408]. This approach initiates the possibility of intraoperative delivery of anti-aggregative drugs, which may prevent α-syn seeding in the graft and increase the chance of recovery. Altogether, significant technological advances in the PD research field, including new DBS devices and software, drug delivery and neuroimaging methods, can vastly improve the quality of therapeutic interventions and, consequently, patients’ conditions.

Lastly, a combination of different strategies, targeting α-syn both directly and indirectly, intracellularly and extracellularly or at different steps of aggregation, might be of particular relevance. For instance, disruption of α-syn oligomers with the concomitant amelioration of ER stress, neuroinflammation and/or apoptosis could vastly enhance neuroprotection in the course of PD. Anti-aggregative agents, by dismantling oligomeric and fibrillar species, may lead to overload of low-molecular forms of α-syn and induce ER stress. ER stress inhibition is essential in restoring disrupted proteostasis and prevention of apoptosis of DA neurons. Also, modulation of UPR and amelioration of ER stress may improve processing of misfolded α-syn. Combination therapy possesses significant advances over single-agent therapy for it increases treatment efficacy, prevents development of drug resistance, reduces drug dosage and duration of treatment and decreases risk of adverse effects. Targeting several molecular events essential to α-syn pathology at the same time may emerge as a new groundbreaking strategy against α-synucleinopathy, which could eventually be implemented into clinical use. We believe that the next decade may include drastic changes to diagnostics and treatment of PD so that the disease finally becomes curable one day.

## Figures and Tables

**Figure 1 pharmaceutics-15-02051-f001:**
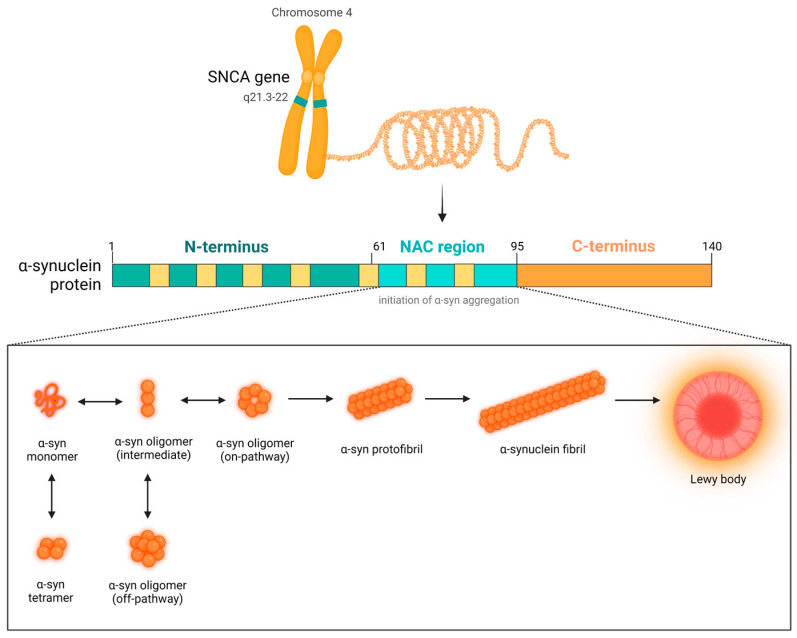
The structure and various strains of α-synuclein (α-syn). α-syn protein, encoded by *SNCA* gene (chromosome 4; q21.3–22), is composed of three domains: N-terminus, NAC (non-amyloid component) and C-terminus. There are 7 “KTKEGV” repeat motifs that run throughout the N-terminal and NAC domains (indicated in yellow). NAC region is responsible for initiation of α-syn aggregation. α-syn monomer can interconvert with other physiological forms (tetramers) or intermediate oligomeric species. Intermediate species may in turn form off-pathway oligomers, which are regarded as toxic, or on-pathway oligomers, which seed further aggregation. On-pathway oligomers assemble into protofibrils, fibrils and Lewy bodies, respectively.

**Figure 2 pharmaceutics-15-02051-f002:**
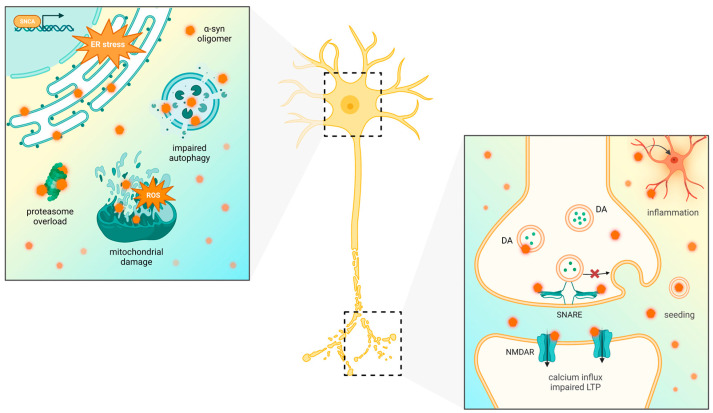
Damage induced by α-synuclein (α-syn) oligomers to the neuronal soma (left panel) and the synapse (right panel). Overexpression of *SNCA* favors aggregation of wild-type α-syn. At the cellular level, oligomers induce endoplasmic reticulum (ER) stress, proteasome overload as well as damage to mitochondria and lysosomes. Oligomers impair synaptic function by interfering with the soluble N-ethylmaleimide-sensitive factor attachment protein receptor (SNARE) complex, decreasing dopamine (DA) release and activating N-methyl-D-aspartate receptors (NMDAR). The latter promotes calcium influx and impairs long-term potentiation (LTP). When toxic species enter the extracellular space, they may seed and propagate to other cells and brain regions, as well as induce neuroinflammatory response in glial cells.

**Table 1 pharmaceutics-15-02051-t001:** Active and passive immunization strategies against α-synuclein (α-syn) in the form of vaccines and monoclonal antibodies, respectively.

**Active Immunization**				
**Target**	**Compound Name**	**Type**	**Current Status**	**Results**
C-terminus (aa 110–130); aggregates	PD01A	α-syn-mimicking peptide with adjuvant	CT phase I; Completed	safe, well-tolerated, substantial immune response in PD [391] and early MSA [392]
C-terminus (aa 110–130); aggregates	PD03A	α-syn-mimicking peptide with adjuvant	CT phase I; Completed	safe, well-tolerated, substantial immune response in early PD and early MSA [393]
C-terminus (10 aa fragment); aggregates	UB-312	α-syn-mimicking peptide with adjuvant	CT phase I; completed (PD) CT phase II; recruiting (PD, MSA)	safe, well-tolerated, substantial immune response in healthy volunteers [394]
C-terminus (aa 126–140); aggregates	α-Syn_126-140_-P30	α-syn-mimicking peptide with adjuvant	Preclinial	substantial immune response in mice [395]
C-terminus (aa 85–99, 109–126, 126–140); aggregates	PV-1950D	α-syn DNA	Preclinical	↓ total and PK-resistant α-syn level in tg mice ↓ neuroinflammation in tg mice [390]
**Passive Immunization**				
**Target**	**Compound Name**	**Type**	**Current Status**	**Results**
C-terminus (aa 120–140); monomers and aggregates	Ab274	Murine IgG2a	Preclinical	↓ α-syn level in cells ↓ neuroinflammation in cells ↑ behavioral and motor function in tg mice [396]
C-terminus (aa 118–126); monomers	9E4, 5C1, 5D12	Murine IgG1	Preclinical	↓ α-syn level (also truncated forms) in cells ↓ neuroinflammation in cells ↑ behavioral and motor function in tg mice [397]
C-terminus (aa 91–99); monomers	1H7			
C-terminus (aa 118–126); aggregates	PRX002/prasinezumab	Human IgG1 (humanized version of 9E4)	CT phase II; active, not recruiting	no efficacy over placebo and safety concerns [398]
C-terminus (aa 127–140); aggregates	Syn-O1 Syn-O2 Syn-O4 Syn-F1 Syn-F2	IgG2b IgG1 IgG1 IgG1 IgG2a	Preclinical	↓ α-syn level ↓ neuroinflammation ↑ behavioral function in tg mice [399]
C-terminus (aa 121–127); soluble aggregates	ABBV-0805/ BAN0805/mAb47	Human IgG1 (humanized version of mAb47)	CT phase I; withdrawn	study withdrawn due to strategic considerations [NCT04127695]
N-terminus (aa 1–10); aggregates	BIIB054/cinpanemab	Human IgG1	CT phase II; discontinued	study did not meet its primary and secondary outcome measures [NCT03318523]
C-terminus (aa 102–130); monomers and aggregates	MEDI1341	Human IgG1	CT phase I; completed (PD) CT phase II; recruiting (MSA)	data unpublished [NCT04449484]
C-terminus (aa 121–125); insoluble aggregates	Syn211	Murine IgG1	Preclinical	↓ α-syn propagation in cells ↓ α-syn oligomer toxicity in cells ↑ motor function in PFF-inoculated mice (Syn303 only) [400]
N-terminus (aa 1–5); insoluble aggregates	Syn303			
N-terminus (aa 16–35); aggregates	AB1	Goat IgG	Preclinical	↓ α-syn levels in tg rats ↓ neuroinflammation in tg rats ↑ behavioral function in tg rats [401]
Unknown; oligomers	syn-D5 syn-10H	scFv	Preclinical	↓ α-syn aggregation in cells ↓ α-syn oligomer toxicity in cells ↑ behavioral function in tg mice [402]
NAC region (aa 71–77); oligomers	W20	scFv	Preclinical	↓ neuroinflammation in tg mice ↑ cognitive and motor function in tg mice [403]

Abbreviations: α-syn—α-synuclein; NAC—non-amyloid component; aa—amino acid residues; CT—clinical trial; PD—Parkinson’s disease; MSA—multiple system atrophy; tg—transgenic; PFF—pre-formed fibrils; PK—proteinase K; ↓—decreased; ↑—increased/improved.

## Data Availability

Not applicable.

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
