# Peer review of "Inhibition of Protein Aggregation and Endoplasmic Reticulum Stress as a Targeted Therapy for α-Synucleinopathy"

_pharmaceutics, 2023, doi:10.3390/pharmaceutics15082051_

Round 1
Reviewer 1 Report
The review highlights the importance of the pathological role of α-synuclein in Parkinson's disease (PD) and the activation of endoplasmic reticulum (ER) stress. It highlights the need for new disease-modifying strategies and explores potential therapeutic approaches targeting α-synuclein and associated pathogenic mechanisms.
Abstract:
1. The statement that α-synucleinopathies are "one of the leading causes of neurodegeneration nowadays" is too broad and needs to be supported by specific statistics. Fully aware of the abstract restrictions
2. Start the abstract with a concise introduction to α-synuclein, its normal function, and its association with neurodegenerative diseases.
3. Elaborate on the potential therapeutic approaches of "anti-aggregative agents" and "natural compounds" by providing specific examples and referencing studies that have explored their effectiveness.
Introduction:
1. The introduction provides a general overview of Parkinson's disease (PD) but lacks a clear statement of the purpose or objective of the review. It would be helpful to explicitly state the aim of the review in terms of what knowledge gaps or questions it intends to address.
2. While the section mentions the abnormal accumulation of α-synuclein in the substantia nigra as the major molecular event underlying PD, it does not provide a comprehensive explanation of α-synuclein's role in disease pathophysiology. Incorporating references to studies that have investigated the mechanisms by which α-synuclein contributes to dopaminergic neuron loss and the progression of PD would strengthen the argument.
3. The physiological role of α-synuclein as a monomer regulating synaptic vesicle trafficking is mentioned but could be further expanded upon. Incorporating references to studies exploring the normal function of α-synuclein and how disruptions in this function contribute to disease pathogenesis would provide a more comprehensive context.
4. The discussion on treatment options for PD is limited to dopamine replacement therapy, and there is no mention of other symptomatic treatments or non-pharmacological approaches. Expanding this section to include a broader range of treatment options and referencing relevant studies or clinical guidelines would provide a more comprehensive overview.
Section 2
5. includes references to several sources (e.g., [33], [34], [35–38]), but these references are not provided in the text or a bibliography at the end. Properly citing the sources would enhance the credibility of the information presented.
6. The section primarily presents factual information about the structure of the SNCA gene and α-synuclein protein, as well as other genes related to α-synuclein pathology. However, it does not offer a critical analysis or interpretation of the information presented. Including critical insights or discussing the significance of the findings would add depth to the section.
7. The section does not mention any recent research or advancements in the field. Given that the knowledge cutoff is September 2021 and the passage does not provide a publication date, it may not include the most up-to-date information. Including recent research findings would make the passage more relevant and current.
Section 3
8. Expand on the role of ER stress and the UPR pathway in PD, discussing their relevance to α-synuclein-induced neurodegeneration in more detail.
9. Elaborate on the functions and interactions of the three major UPR sensors (IRE1α, PERK, and ATF6) and their specific involvement in α-synuclein-induced ER stress. The review is already long and a paragraph wont make a difference
10. Discuss the importance of maintaining a proper balance of PERK activation and its implications for proteostasis and neuroprotection, providing relevant studies that support this claim
11. The text lacks a cohesive synthesis of the discussed approaches and their overall implications for the treatment of α-synucleinopathy. A clear conclusion summarizing the main points and highlighting the most promising or impactful strategies would provide a more comprehensive and satisfying review.
12. The challenges in developing pharmacological modulators for α-syn are mentioned, but no suggestions or references to relevant literature are provided to address these challenges.
13. The discussion on unidentified pathogenic forms of α-syn and the trigger for α-syn misfolding lacks supporting evidence or references to studies exploring these aspects.
14. The mention of non-invasive drug delivery, targeting α-syn in the enteric nervous system (ENS), and advances in induced pluripotent stem cells (iPSCs) could benefit from specific examples, mechanisms of action, and supporting references.
15. The argument for combination therapy lacks specific evidence or references to studies demonstrating the enhanced efficacy of such approaches
16. The mention of a biomarker for α-syn from the Parkinson's Progression Markers Initiative (PPMI) study is interesting, but it would be beneficial to provide more details and references to the relevant study.
Reviewer 2 Report
In this review, the authors discussed the structures/forms of α-Synuclein, endoplasmic reticulum stress (ERS)/other molecular events induced by α-Synuclein, as well as drugs/strategies targeting α-Synuclein. The article includes lots of information (488 references) and important updates. A few concerns below may worth consideration to further improve the manuscript.
1. I would suggest a concise rather than detailed description of structures α-Synuclein. However, the strains could be better described since recent publications demonstrated the disease-associated/distinct α-Synuclein aggregates (Nature 2020;578:273-7; Neurology 2022;99:e2417-e27.)
2. In the ERS/other molecular events, I would suggest an emphasis on propagated α-Synuclein. For example, extracellular α-Synuclein could induce ERS (Neurochem Res. 2021). The effects of overexpressed/aggregated α-Synuclein have been reviewed extensively in previous publications.
3. The title focused on ERS as a targeted therapy, while there is much more than ERS in the article. Please consider the consistency between the title and main text.
4. “Other Genes Related to α-Synuclein Pathology” seems redundant. I understand the gene are related to α-Synuclein, and I would suggest an emphasis on those related to both ERS and α-Synuclein. The other sections of the review need more focus on ERS.
5. The section“Drugs Targeting α-Synuclein” is informative and well-organized. Is it possible to incorporate “mitochondria””metal”into other parts?
Round 2
Reviewer 1 Report
No Further comments!
Reviewer 2 Report
NA